# CATALOG-NATIVE LLM: SPEAKING ITEM-ID DIALECT WITH LESS ENTANGLEMENT FOR RECOMMENDATION

**Reza Shirkavand**[*1], **Xiaokai Wei**[2], **Chen Wang**[2], **Zheng Hui**[*3], **Heng Huang**[1] , **Michelle Gong**[2]
[1]University of Maryland - College Park {`rezashkv,heng`}`@cs.umd.edu`,
[2]Roblox {`xwei,cwang,mgong`}`@roblox.com`
[3]University of Cambridge `zh2483@columbia.edu`

## ABSTRACT

While collaborative filtering delivers predictive accuracy and efficiency, and Large Language Models (LLMs) enable expressive and generalizable reasoning, modern recommendation systems must bring these strengths together. Growing user expectations, such as natural-language queries and transparent explanations, further highlight the need for a unified approach. However, doing so is nontrivial. Collaborative signals are often token-efficient but semantically opaque, while LLMs are semantically rich but struggle to model implicit user preferences when trained only on textual inputs. This paper introduces Item-ID + Natural-language Mixture-of-Experts Language Model (IDIOMoE), which treats item interaction histories as a native dialect within the language space, enabling collaborative signals to be understood in the same way as natural language. By splitting the Feed Forward Network of each block of a pretrained LLM into a separate text expert and an item expert with token-type gating, our method avoids destructive interference between text and catalog modalities. IDIOMoE demonstrates strong recommendation performance across both public and proprietary datasets, while preserving the text understanding of the pretrained model.

## 1 INTRODUCTION

Recommendation systems shape what people read, watch, buy, learn, and play. As AI shifts from static predictors to reasoning agents capable of following instructions, recommendation is also evolving from ranking fixed lists to assisting users in exploring, planning, and deciding. This trend is visible in practice: Amazon's Rufus provides LLM-powered conversational shopping (Amazon, 2024); Meta's Llama-3 assistant is embedded in WhatsApp, Instagram, and Facebook for task planning (Meta, 2024); and Netflix is adopting foundation-model approaches for personalization and LLM-based conversational retrieval (Netflix, 2025; Zhu et al., 2025). These examples motivate bringing LLM knowledge and instruction-following into recommenders while preserving the collaborative patterns that make them accurate at scale.

Conventional recommenders like collaborative filtering (CF)(Koren et al., 2009), content-based (CB)(Lops et al., 2011), and sequential models (Kang & McAuley, 2018; Sun et al., 2019; Zhai et al., 2024) perform well within their scope when data are abundant, but they depend heavily on the quality of logs and item attributes. They remain vulnerable to popularity bias (Abdollahpouri et al., 2019), struggle to integrate heterogeneous signals (text, behavior, and context), and cannot support natural language queries.

Pre-trained LLMs offer complementary strengths: they bring broad world knowledge, can follow natural-language instructions, and can reason about multi-objective trade-offs. Yet a fundamental gap remains. LLM pretraining centers on semantic understanding, whereas recommendation requires modeling collaborative preference patterns. The key challenge is leveraging LLMs for preference understanding without disrupting their semantic competence.

Recent work has tried to bridge this gap by extending LLM vocabularies with item IDs (Cao et al., 2024; Zhu et al., 2024; Jiang et al., 2025; Zhang et al., 2025), enabling direct ID-level generation.

---

*Work done during internship at Roblox

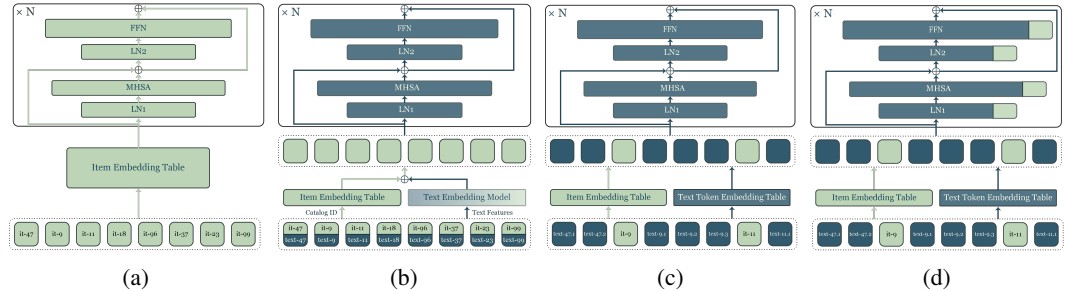

Figure 1: Four designs for recommendation with Transformers/LLMs. (a) ID-only Transformer: trained from scratch on item-ID sequences, with no pretrained LLM involved. (b) Text-derived bias: a pretrained LLM on IDs, with an external text encoder providing side features that bias item scores. (c) Explicit text tokens: a pretrained LLM that directly consumes both item-ID tokens and (possibly) text tokens in the same sequence. (d) Explicit text tokens + extra capacity: like (c), but adds item-specific parameters to better handle IDs. IDIOMoE is a special case of (d).

While effective in principle, such naive integration often causes knowledge interference: collaborative signals entangle with linguistic semantics, leading to degraded performance on both sides. As we'll show, this interference does not vanish by simply scaling up parameters (e.g. adding more parameters naively) and thus calls for more principled architectural solutions.

Inspired by mixture-of-experts (MoE) (Shazeer et al., 2017; Lepikhin et al., 2020; Fedus et al., 2022), we view ItemID modeling as a dialect distinct from natural language. But unlike standard MoE, which routes tokens indiscriminately, we design a targeted *Item-ID + Natural-language Mixture-of-Experts Language Model (IDIOMoE)* that assigns a dedicated collaborative expert for IDs alongside a preserved text expert for language. A token-type gate orchestrates their interaction, mitigating interference while retaining pretraining knowledge. Evaluations on both public benchmarks and a real-world industrial dataset from a leading online platform with hundreds of millions of users show that IDIOMoE consistently outperforms text-only adapters and item-only baselines. Our main contributions are:

**Disentangled MoE architecture for recommendation.**  We propose a Mixture-of-Experts design that treats Item-IDs as a native dialect. To the best of our knowledge, this is the first attempt at separating collaborative filtering from semantic processing, with a router that activates text experts only when useful.

**Robust performance on real-world scale.**  Our method achieves compelling results on public datasets and on our large proprietary dataset with more hundreds of millions of users, while maintaining the natural language understanding of a pre-trained LLM.

**Extensive ablations isolating the source of gains.**  We study model capacity and matched-capacity non-MoE baselines showing that improvements arise from expert specialization and routing, not just added parameters.

**Analysis of expert specialization.**  Through a key-value memory lens of FFN neurons, we show that MoE separation yields clearer item-text affinity, higher category purity, and more clustered neurons than a non-MoE baseline, providing evidence that expert disentanglement leads to more interpretable and modular representations.

## 2 RELATED WORK

### 2.1 CONVENTIONAL RECOMMENDATION METHODS

Traditional recommendation models fall into collaborative filtering (CF), content-based (CB), and sequential paradigms. CF learns from user–item interactions to model latent preferences (Koren et al., 2009), while CB leverages item attributes to improve personalization and mitigate cold-start

issues (Lops et al., 2011). Sequential models further capture temporal dynamics, using models such as RNNs (Hidasi et al., 2015), SASRec (Kang & McAuley, 2018), and BERT4Rec (Sun et al., 2019). Though these models achieve strong performance under sufficient data, they operate on opaque ID sequences and require hand-crafted features or specialized architectures to incorporate diverse signals like language or intent. They also struggle with long-tail exposure (Abdollahpouri et al., 2019).

## 2.2 GENERATIVE RECOMMENDATION

Some works treat recommendation as sequence generation, unifying retrieval and ranking under a generative objective (Yang et al., 2025). This includes large-scale decoder models such as HSTU (Zhai et al., 2024), which scales to trillions of parameters, and OneRec (Deng et al., 2025b), which uses a sparse MoE encoder–decoder architecture for scalable training. These approaches improve novelty, fluency, and explainability, but are resource-intensive and require careful objective and data design to fully exploit collaborative interaction signals. They also do not support conversational recommendation.

### 2.2.1 LLM-BASED RECOMMENDATION AND SEMANTIC–ID ALIGNMENT

Large language models (LLMs) offer world knowledge and instruction-following capabilities that are appealing for building explainable recommenders. Recent frameworks such as P5 (Geng et al., 2022) reframe recommendation tasks as text-to-text generation, supporting few-shot generalization. Prompt-based methods (Hou et al., 2024b) further explore LLMs as zero-shot rankers. However, these methods require verbose text inputs and often discard raw user–item interaction data, missing collaborative patterns entirely. To bridge this semantic collaborative gap, prior work fine tunes on interactions (Cao et al., 2024), aligns with rewards (Lu et al., 2024), or unifies modalities in shared token spaces (Zhai et al., 2025). A complementary direction embeds item IDs as tokens (e.g., CoVE (Zhang et al., 2025), CLLM4Rec (Zhu et al., 2024), URM (Jiang et al., 2025)), enabling token efficient generation and retrieval. However, designs like URM that drop explicit text tokens, hinder conversational recommendation and instruction handling. And when ID tokens and text tokens share parameters, interference emerges: language and collaborative signals entangle, degrading both.

## 2.3 MULTIMODAL MoE LLMS

Recent work integrates MoE into multimodal LLMs (MLLMs) and LVLMs Bao et al. (2022); Shen et al. (2023); Diao et al. (2025); Deng et al. (2025a). MoE-LLaVA (Lin et al., 2024a) adds a sparse MoE backbone to LLaVA (Liu et al., 2023a), converting feed-forward blocks into experts to match or exceed larger dense variants while activating fewer parameters and reducing visual hallucinations. Uni-MoE (Li et al., 2025) scales unified multimodal LLMs across many modalities and tasks with MoE layers. MoME (Xu et al., 2024) further mitigates task interference by factorizing the model into a Mixture of Vision Experts (MoVE) and a Mixture of Language Experts (MoLE), with MoVE aggregating multi-encoder vision features via an instruction-conditioned router and MoLE using sparsely gated adapter experts.

## 2.4 MOTIVATION AND POSITIONING

While prior work has shown the potential of combining semantic understanding with collaborative signals, existing methods lack clear mechanisms to separate and preserve these distinct forms of knowledge. Text can be incorporated via (a) *text-as-features* (pre-encoded embeddings/biases attached to IDs); or (b) *explicit text tokens* (Figure 1). We choose the latter to preserve conversational capabilities of the LLM. In this setting, interference between language understanding and ID-level preference modeling remains an underexplored bottleneck. Simply mixing tokens or scaling capacity does not solve it.

We address this challenge by introducing a *Item-ID + Natural-language Mixture-of-Experts Language Model (IDIOMoE)* that treats item interactions as a native dialect. IDIOMoE dedicates separate pathways to item and text processing in each block, with a lightweight token-type gate that reduces interference while retaining language understanding. This design enables efficient ID-level modeling and better alignment with both semantic and collaborative objectives.

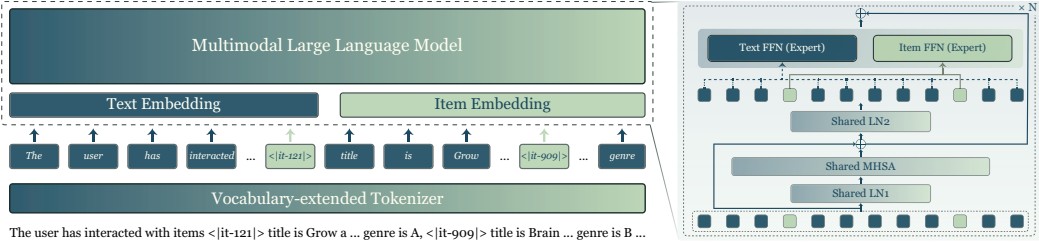

Figure 2: Overview of our proposed IDIOMoE. We extend the LLM tokenizer with new `"item-id"` tokens and introduce a dedicated item embedding layer. The Normalization and Attention layers are shared across all token types, while tokens are routed to distinct FFN layers depending on their type.

Table 1: Improvements over the ID-only baseline when adding text features.

| Variant | Arts Δ(%) | | Industrial Δ(%) | |
|---|---|---|---|---|
| | HR@10 | NDCG@10 | HR@10 | NDCG@10 |
| ID-only (baseline) | — | | | |
| ID-only + text-derived bias | +42.8% | +26.4% | +18.1% | +13.9% |
| ID + explicit attributes | +24.6% | +17.6% | +11.4% | +6.8% |
| **IDIOMoE** | **+44.1%** | **+28.1%** | **+22.7%** | **+14.2%** |

Figure 3: Language understanding retention.

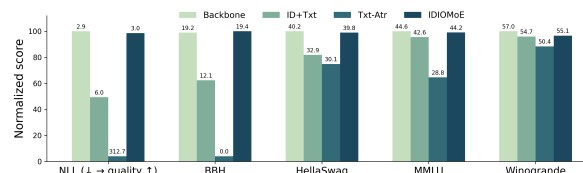

# 3 METHOD

## 3.1 PRELIMINARY

We study how incorporating item textual attributes affects performance given a user's interaction history. We start from the pretrained `Qwen/Qwen2.5-0.5B` (Qwen et al., 2025), extend its vocabulary with item-ID tokens, and compare three variants that differ only in input format and the source of item embeddings. In all variants, instruction text tokens are embedded with the LLM's native token embedding matrix.

1. **ID-only (learned ID embeddings).** Input: *"The user has interacted with `<|item-53|>` `<|item-11|>`..."*. Each item token is embedded via a learned item embedding table.

2. **ID-only + text-derived bias.** Following Jiang et al. (2025) this variant has same input as (a). However, each item token embedding is the sum of (i) a learned ID vector and (ii) a text-derived vector computed from the item's title and category using a general-purpose sentence-embedding model.

3. **ID + explicit attributes.** Input interleaves IDs with attributes: *"The user has interacted with `<|item-53|>` title: X, category: Y; `<|item-11|>`..."*. Item-ID tokens use the learned item embedding table; Text tokens are embedded by the LLM's token embeddings.

We evaluate the above on two datasets: Amazon-Arts (Ni et al., 2019) and our industrial dataset. The results are presented in Table 1. In both datasets adding item textual attributes improves performance. The text-derived bias approach performs better as it is easier for the model to handle as it adds some semantic signal without making the sequence longer or more complex. In contrast, giving the model full attribute text makes the input longer and harder to learn from. But there is a key reason to still include explicit text: it enables capabilities that the bias method can't. Conversing with users and generating user-friendly explanations all rely on having real text.

To evaluate whether the variants preserve the pretrained model's linguistic ability, we measure negative log-likelihood (NLL) on 5,000 samples from the `wikitext` validation set (Merity et al., 2016) and further assess performance on four benchmarks: BBH (Suzgun et al., 2022), HellaSwag (Zellers et al., 2019), MMLU (Hendrycks et al., 2021), and WinoGrande (Sakaguchi et al., 2019). As shown in Figure 3, ID+Text achieves substantially lower NLL and significantly higher benchmark results compared to the text-derived bias variant. While the bias method provides strong recommendation accuracy, it does so at the cost of language degradation, reflected in much poorer performance on

language understanding tasks. This points to the need for a better approach; one that preserves the advantages of explicit text for conversational recommendation while still achieving strong performance on standard recommendation tasks.

In this paper, we propose to divide responsibilities rather than forcing a single model to handle everything. One expert is dedicated to IDs and collaborative filtering, while another is responsible for text. This design allows us to retain the benefits of explicit text when needed, without sacrificing efficiency or accuracy when it is not. We show IDIOMoE preserves the language understanding of the model, while delivering the best recommendation performance (Table 1 and Figure 3), confirming that separating experts by token type reduces semantic–collaborative interference.

## 3.2 IDIOMoE

We present the *Item-ID + Natural-language Mixture-of-Experts Language Model* (IDIOMoE), a pretrained decoder-only LLM augmented with item-specialized experts and native item tokens. IDIOMoE keeps the language skills of the base model intact while learning collaborative patterns directly from user-item sequences. We start from a pretrained causal transformer and replace each feed-forward network (FFN) with a two-expert module:

- **Text Expert**: the original FFN from the pretrained LLM, preserved as-is.
- **Item Expert**: a new FFN similar to the text expert, optionally shrunk (e.g., $\times\frac{1}{2}$, $\times\frac{1}{4}$) to add capacity efficiently.

Routing is handled by a **static token-type gate**: We use a simple static routing scheme: only item-ID tokens `<|it-.|>` are routed to the item expert, and all other tokens (titles, attributes, etc.) are routed to the text expert. All tokens share the same self-attention layers at every depth, so IDs and text always attend to each other, and the MoE split only affects the FFN sublayers, i.e., where ID- vs. text-specific information is stored. This design lets the model jointly reason over blended textual attributes and item IDs while allocating separate capacity for catalog structure. Moreover, since only one expert is active per token, so compute stays comparable to the base model (See Appendix B.6.4 for a discussion of efficiency results). Figure 2 provides an overview of our framework.

### 3.2.1 Native item tokens and hybrid head.

We augment the tokenizer with special item tokens `<|it-id|>` and attach a hybrid embedding layer that combines the frozen text embeddings with a trainable item embedding table. The output head reuses the same hybrid parameterization so the model can generate item IDs directly.

## 3.3 FFN Key-Value Memory Analysis

### 3.3.1 Setup

Following Geva et al. (2022), we view each feed-forward network (FFN) in a transformer block as a key-value memory, where hidden states act as queries and FFN neurons contribute value vectors. Our goal is to probe whether Mixture-of-Experts (MoE) separation encourages the *item expert* to encode item semantics distinct from the *text expert*, and how this differs from a non-MoE baseline.

For a transformer layer $\ell \in \{1, \ldots, L\}$, let the FFN consist of two linear projections with activation in between. We denote the second projection as $W_{\text{out}}^{(\ell)} \in \mathbb{R}^{I \times d}$ where $I$ is the FFN hidden dimension and $d$ is the model dimension. Each row $w_j^{(\ell)} \in \mathbb{R}^d$ of $W_{\text{out}}^{(\ell)}$ is treated as a *value vector* associated with neuron $j$ in layer $\ell$. To study how these rows align with model embeddings, we construct two sets of reference vectors:

- **Item embeddings:** $E_{\text{items}} \in \mathbb{R}^{N_{\text{items}} \times d}$, taken from the learned item embedding table used for ID tokens.
- **Text token embeddings:** $E_{\text{text}} \in \mathbb{R}^{V_{\text{text}} \times d}$, taken from the backbone's input embedding matrix for standard vocabulary tokens (excluding items).

Table 2: Results on small Amazon catalogs. Highlight = LLM-Based. Bold = best; underline = second best; "–" = unreported. [1] Zhai et al. (2025). [2] Cao et al. (2024). [3] Zhang et al. (2025).

| Method | Games | | Instruments | | Arts | | Sports | | Beauty | | Toys | |
|---|---|---|---|---|---|---|---|---|---|---|---|---|
| | NDCG@10 | HR@10 | NDCG@10 | HR@10 | NDCG@10 | HR@10 | NDCG@10 | HR@10 | NDCG@10 | HR@10 | NDCG@10 | HR@10 |
| GRU4Rec[1,2] | 0.0453 | 0.0895 | 0.0857 | 0.1207 | 0.0690 | 0.1088 | 0.0110 | 0.0204 | 0.0137 | 0.0283 | 0.0084 | 0.0176 |
| Bert4Rec[1,2] | 0.0366 | 0.0725 | 0.0739 | 0.1081 | 0.0575 | 0.0922 | 0.0099 | 0.0191 | 0.0170 | 0.0347 | 0.0099 | 0.0203 |
| FDSA[1,2] | 0.0509 | 0.0988 | 0.0859 | 0.1249 | 0.0695 | 0.1190 | 0.0156 | 0.0288 | 0.0208 | 0.0407 | 0.0189 | 0.0381 |
| S3-Rec[1,2] | 0.0468 | 0.0903 | 0.0743 | 0.1123 | 0.0630 | 0.1030 | 0.0240 | 0.0385 | 0.0327 | 0.0647 | 0.0376 | 0.0700 |
| TIGER[1,2] | 0.0453 | 0.0857 | 0.0950 | 0.1221 | 0.0806 | 0.1167 | 0.0225 | 0.0400 | 0.0384 | 0.0648 | 0.0432 | 0.0712 |
| VQ-Rec[1] | 0.0329 | 0.0679 | 0.0891 | 0.1357 | 0.0844 | _0.1386_ | - | - | - | - | - | - |
| MISSRec[1] | 0.0499 | 0.1048 | 0.0880 | 0.1361 | 0.0815 | 0.1321 | - | - | - | - | - | - |
| P5-CID[1] | 0.0454 | 0.0824 | 0.0704 | 0.1119 | 0.0662 | 0.0994 | - | - | - | - | - | - |
| VIP5[1] | 0.0418 | 0.0758 | 0.0872 | 0.1071 | 0.0635 | 0.0859 | - | - | - | - | - | - |
| MQL4GRec[1] | 0.0548 | 0.1033 | **0.1060** | _0.1375_ | _0.0950_ | 0.1327 | - | - | - | - | - | - |
| ReAT[2] | - | - | - | - | - | - | 0.0232 | 0.0422 | 0.0535 | 0.0722 | 0.0461 | 0.0776 |
| E4SRec[2] | - | - | - | - | - | - | 0.0237 | 0.0410 | 0.0435 | 0.0758 | 0.0479 | 0.0798 |
| IDGenRec[2] | - | - | - | - | - | - | _0.0372_ | 0.0574 | 0.0541 | 0.0814 | _0.0551_ | 0.0870 |
| CoVE[3] | - | - | - | - | - | - | 0.0359 | _0.0624_ | _0.0593_ | 0.1009 | **0.0595** | **0.0986** |
| SASRec | 0.0547 | 0.0997 | 0.0749 | 0.1256 | 0.0927 | 0.1290 | 0.0289 | 0.0531 | 0.0541 | _0.0945_ | 0.0542 | _0.0958_ |
| HSTU | **0.0609** | _0.1089_ | 0.0712 | 0.1214 | 0.0941 | 0.1301 | 0.0287 | 0.0515 | 0.0474 | 0.0863 | 0.0536 | 0.0933 |
| ID Transformer | 0.0392 | 0.0669 | 0.0709 | 0.0761 | 0.0824 | 0.1025 | 0.0081 | 0.0122 | 0.0314 | 0.0503 | 0.0271 | 0.0405 |
| Text-Attr LLM | 0.0464 | 0.0862 | 0.0778 | 0.1133 | 0.0938 | 0.1374 | 0.0251 | 0.0497 | 0.0390 | 0.0761 | 0.0502 | 0.0895 |
| Item-LLM | 0.0407 | 0.0734 | 0.0943 | 0.1095 | 0.0901 | 0.1272 | 0.0211 | 0.0369 | 0.0449 | 0.0738 | 0.0410 | 0.0704 |
| **IDIOMoE** | _0.0605_ | **0.1102** | _0.1054_ | **0.1385** | **0.1029** | **0.1409** | **0.0391** | **0.0674** | **0.0665** | **0.1104** | 0.0531 | 0.0927 |

Given a value vector $w \in \mathbb{R}^d$, we compute cosine similarities to both sets:

$$s_{\text{items}}(w) = E_{\text{items}}w^\top, \quad s_{\text{text}}(w) = E_{\text{text}}w^\top, \tag{1}$$

assuming all vectors are $\ell_2$-normalized. We then retrieve the top-$k$ most similar item IDs and text tokens for analysis.

### 3.3.2 METRICS

We define three metrics to quantify the specialization of each neuron's value vector $w$:

$$\textbf{Affinity:} \quad a(w) = \text{median}\big(s_{\text{items}}^{\text{top-}k}(w)\big) - \text{median}\big(s_{\text{text}}^{\text{top-}k}(w)\big), \tag{2}$$

$$\textbf{Purity:} \quad p(w) = \max_{c \in \mathcal{C}} \frac{1}{k} \big| \{ i \in \text{top-}k(w) : \text{cat}(i) = c \} \big| \in [0, 1], \tag{3}$$

$$\textbf{Clustered row:} \quad \mathbf{1}_{\text{cluster}}(w) = \mathbb{I}\big[p(w) \geq \tau\big], \quad \text{for threshold } \tau \in [0, 1]. \tag{4}$$

Here, $\mathcal{C}$ denotes the set of item categories, $\text{cat}(i)$ returns the category of item $i$, and $\tau$ controls the strictness of cluster assignment. In simple terms, affinity quantifies the relative alignment of an FFN neuron's value vector with item versus text embeddings, thereby indicating modality preference. Purity measures the concentration of a neuron's top-$k$ nearest neighbors within a single item category, reflecting category-specific specialization. Clustered rows are those neurons whose purity exceeds a threshold $\tau$, identifying dimensions of the FFN value space that form coherent category-level clusters.

## 4 EXPERIMENTS

### 4.1 EXPERIMENTAL SETTINGS

**Baselines** Our main focus is on LLM-based recommenders, so the most relevant baselines are different ways of adding recommendation capability to LLMs. We include established LLM-for-Rec baselines that are directly comparable to our setting: the P5/P5-CID family, which reframes recommendation as text-to-text generation over a pretrained language model (Geng et al., 2022; Hua et al., 2023); VIP5, a multimodal extension of P5 that adapts the LLM with parameter-efficient modules (Geng et al., 2023); E4SRec, which keeps the LLM largely frozen and adds a lightweight ID-side adapter for sequential recommendation (Li et al., 2023d); and ReAT, which aligns LLMs to recommendation objectives via auxiliary, recommendation-specific generated tasks (Cao et al., 2024). These capture the main design choices for adding recommendation capability to LLMs (prompting, adapters, frozen-backbone adapters, alignment), and thus form our most relevant comparison set. In

Table 3: Results on large Amazon catalogs. Bold=best; underline=second best; Highlight=LLM-Based

| Method | Beauty | | Books | | Toys | |
|---|---|---|---|---|---|---|
| | NDCG@10 | HR@10 | NDCG@10 | HR@10 | NDCG@10 | HR@10 |
| SASRec | 0.0051 | 0.0101 | 0.0064 | 0.0128 | 0.0122 | 0.0245 |
| HSTU | **0.0130** | **0.0247** | 0.0211 | 0.0410 | 0.0149 | 0.0332 |
| ID Transformer | 0.0068 | 0.0095 | **0.0224** | 0.0295 | 0.0048 | 0.0079 |
| Text-Attr LLM | 0.0105 | 0.0163 | 0.0195 | 0.0290 | 0.0164 | 0.0300 |
| Item-LLM | 0.0082 | 0.0119 | 0.0174 | 0.0261 | 0.0079 | 0.0148 |
| IDIOMoE | 0.0119 | 0.0228 | **0.0224** | **0.0419** | **0.0186** | **0.0361** |

Figure 4: Results on our industrial dataset.

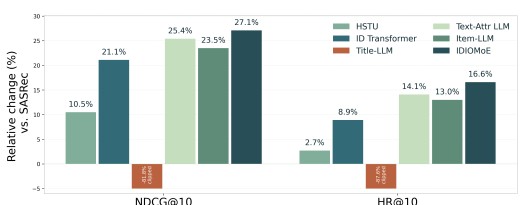

addition, we compare three variants built on the same backbone: (i) *ID Transformer*, trained only on item tokens; (ii) *Item-ID LLM + text-derived bias* (Jiang et al., 2025), where ID embeddings are augmented with text features; and (iii) *Item-LLM*, which integrates item text via vocabulary expansion but without MoE. These three variants are matched to IDIOMoE in parameter count and trained under identical token budgets. For completeness, we also report results of classical sequence models (GRU4Rec (Hidasi et al., 2015), Bert4Rec (Sun et al., 2019), FDSA (Zhang et al., 2019), S3-Rec (Zhou et al., 2020)), recent quantized/contrastive approaches (VQ-Rec (Hou et al., 2023b), MissRec (Wang et al., 2023a), TIGER (Rajput et al., 2023), MQL4GRec (Zhai et al., 2025), IDGenRec (Tan et al., 2024)), and strong transformer baselines (SASRec (Kang & McAuley, 2018), HSTU (Zhai et al., 2024)). We further include CoVE (Zhang et al., 2025), which extends an LLM with LoRA parameters to encode catalog items. While these embedding-driven or classical models are not our primary comparison targets, we include them for completeness on smaller Amazon datasets. Full baseline details are in Appendix B.1.

**Datasets, Evaluation, & Backbone** We use public Amazon Dataset: Games, Instruments and Arts (Ni et al., 2019) as well as Sports, Beauty and Toys McAuley et al. (2015). We further report performance on larger 2023 Amazon variants (Beauty, Books, and Toys) with substantially larger item vocabularies Hou et al. (2024a). We also train and evaluate on our in-house industrial-scale dataset with hundreds of millions of users and tens of thousands of items. We report NDCG@10, HR@10 and MRR. Metrics are computed over the full catalog on Amazon datasets and on 50000 samples in our industrial dataset. We follow the standard leave last item out procedure for separating train and test datasets. All LLM-based models that we train, use `Qwen/Qwen2.5-0.5B` on text-analysis results, Amazon datasets, and for all ablations. We use `Qwen/Qwen2.5-1.5B` for main results on our proprietary dataset. See Appendix B for all details.

### 4.1.1 RESULTS: AMAZON CATALOGS

Table 2 summarizes performance across six small Amazon datasets. We observe that classical sequence models such as GRU4Rec (Hidasi et al., 2015) and Bert4Rec Sun et al. (2019) perform consistently worse than more recent architectures, confirming the difficulty of modeling sparse item interactions in these settings. Transformer-based methods with additional inductive biases, such as FDSA (Zhang et al., 2019), S3-Rec Zhou et al. (2020), and TIGER Rajput et al. (2023), provide moderate gains, while recent quantization and multi-modal approaches like VQ-Rec Hou et al. (2023b), MISSRec Wang et al. (2023a), and MQL4GRec Zhai et al. (2025) achieve stronger results. Compared to direct LLM-Based baselines (highlighted in gray) and classical sequence models, IDIOMoE delivers the most consistent improvements: it achieves the highest NDCG@10 and HR@10 in nearly all domains. These results highlight the robustness of our approach across diverse catalog sizes and domains, suggesting better generalization than prior methods that either overfit to specific datasets or fail to transfer across settings.

We evaluate SASRec (Kang & McAuley, 2018), HSTU (Zhai et al., 2024), ID-Transformer, LLM-based baselines and IDIOMoE on Larger Amazon datasets. Table 3 presents the results. IDIOMoE is the strongest LLM-based method across all three catalogs: it is the top LLM on Beauty (2nd overall behind HSTU by a small margin), and it achieves the best overall scores on Books and Toys. In contrast, Item-LLM and Text-Attr LLM Jiang et al. (2025) lag behind IDIOMoE across metrics, indicating that simply mixing item/text tokens or adding text-derived biases is insufficient. Together, these results support our claim that disentangling item and language pathways yields higher ranking quality than prior LLM baselines while remaining competitive with the best non-LLM models.

### 4.1.2 RESULTS: PROPRIETARY DATASET

While results on the Amazon datasets remain a useful reference point, we acknowledge their limitations. The benchmarks are relatively small and may contain overlaps that make them easier than real-world scenarios. Therefore, although we report results on these datasets for comparability with prior work, we place greater weight on evaluations conducted on our large-scale proprietary dataset, which we consider a more realistic and meaningful test of recommendation quality.

Figure 4 (Table 9) shows results on our large-scale proprietary dataset as improvements over the SASRec Kang & McAuley (2018) baseline. ID-Transformer achieves strong gains, confirming that transformers can effectively capture collaborative filtering signals when grounded in IDs and given enough compute. Title-LLM, which relies solely on free-form item titles, collapses in performance, highlighting the limitations of text-only representations for recommendation. Item-LLM combines IDs with textual features and yields further improvements, particularly on HR@10, demonstrating the value of jointly modeling collaborative and semantic signals. HSTU Zhai et al. (2024) provides modest gains but falls short compared to the LLM-based approaches and doesn't support explainable recommendation. Finally, our method (IDIOMoE) achieves the largest improvements across all metrics (+27.1% NDCG@10, +16.6% HR@10, +31.2% MRR), showing that disentangling ID and text processing with specialized experts and routing not only preserves collaborative filtering strength but also better leverages semantic cues for robust large-scale recommendation.

## 4.2 ABLATIONS

### 4.2.1 NON-MOE CAPACITY CONTROLS.

To ensure that the improvements of IDIOMoE are not simply due to added parameters, we compare against non-MoE variants with matched capacity. Specifically, we consider three settings: (i) *wide-FFN*, where the feed-forward layers of the transformer blocks are widened to match IDIOMoE 's parameter count; (ii) *append-blocks*, where additional transformer layers are added after the original stack; and (iii) *prepend-blocks*, where extra layers are inserted before the original stack. All models are trained under the same setup as IDIOMoE with the hyperparameters and the same FLOPS. We also compare against a LoRA Hu et al. (2022) variant where low-rank adapters are added across all layers. Table 4 summarizes the results.

We find that simply adding parameters in non-structured ways is insufficient. Wide-FFN improves performance on Amazon-Beauty but only marginally helps in the industrial setting. In contrast, append-blocks and prepend-blocks severely degrade performance across both datasets, likely due to disruption of pre-trained representations or training instability. LoRA-LLM, where low-rank adapters are added across all layers, helps slightly on Amazon-Beauty but fails drastically on the industrial benchmark, highlighting its sensitivity to scale and signal sparsity.

We also compare with various MoE designs. Both MoA (expert attention modules) and MoT (expertized full transformer blocks with cross

Table 4: Non-MoE capacity controls on Amazon-Beauty and Industrial datasets. All variants are matched to IDIOMoE in parameter count. Results are shown as relative improvements over Item-LLM.

| Method | Amazon-Beauty Δ(%) | | Industrial Δ(%) | |
|---|---|---|---|---|
| | NDCG@10 | HR@10 | NDCG@10 | HR@10 |
| Item-LLM (baseline) | — | | — | |
| LoRA-LLM | +21.5% | +7.9% | -79.1% | -76.3% |
| Wide-FFN | +27.0% | +24.9% | +3.8% | +1.3% |
| Append-blocks | -87.8% | -90.3% | -5.5% | -5.3% |
| Prepend-blocks | -97.2% | -95.9% | -15.3% | -16.2% |
| MoA | +48.3% | +46.2% | +20.9% | +27.1% |
| MoT | **+49.3%** | **+51.1%** | +22.5% | +24.8% |
| IDIOMoE | +48.1% | +49.6% | **+24.1%** | **+28.9%** |

attention) yield large improvements over all non-MoE controls. Importantly, IDIOMoE performs on par or better than both, despite using a simpler and more efficient expert design focused solely on FFNs with static routing. Although MoA and MoT are competitive on Amazon-Beauty and occasionally match or slightly exceed IDIOMoE there, we emphasize the industrial-scale results as our primary evidence. On this large setting, the FFN-based MoE of IDIOMoE consistently outperforms MoA/MoT variants. Nonetheless, the pattern we observe might be dataset-dependent. The core idea of IDIOMoE is to treat catalog items as first-class citizens and to separate where information about IDs and text is stored. All three MoE variants we ablate are consistent with this idea. Our choice to place MoE in the FFNs is guided by the stronger and more stable gains we see on the large-scale industrial dataset.

These results confirm that IDIOMoE's performance is not due to raw parameter count, but rather due to its intentional separation of item and language processing via token-type MoE routing. Compared to generic scaling or lightweight tuning (e.g., LoRA), the structured, disentangled pathways in IDIOMoE yield higher accuracy, especially in large-scale settings where interference between item IDs and natural language is more pronounced.

### 4.2.2 ITEM EXPERT CAPACITY

We vary the intermediate width of the item expert per layer by applying different shrink factors to the middle layer of the item FFN experts. Larger shrink factors reduce the parameter count and latency, but they also constrain the model's ability to capture rich collaborative signals. Table 5 presents the results. On Amazon-Beauty, we see that moderate shrink values (2 and 4) provide substantial improvements over the baseline, with shrink=4 yielding the best balance of capacity and efficiency (+41.8% NDCG@10, +26.6% HR@10). However, very aggressive shrinking (shrink=8) reduces gains, suggesting that the item expert becomes under-parameterized. In contrast, results on the industrial dataset show a different trend: shrinking consistently hurts performance, with small but steady drops in both NDCG@10 and HR@10 as capacity decreases.

These findings indicate that while smaller benchmarks can benefit from lighter experts, large-scale real-world data demands higher item-expert capacity to preserve recommendation accuracy. This motivates the need for adaptive capacity allocation, where expert width can be tuned to match the complexity and scale of the target domain. Our method provides this control on capacity allocation.

Table 5: Impact of varying item expert capacity.

| Shrink | Amazon-Beauty $\Delta(\%)$ | | Industrial $\Delta(\%)$ | |
|---|---|---|---|---|
| | NDCG@10 | HR@10 | NDCG@10 | HR@10 |
| 1 (baseline) | — | | — | |
| 2 | +21.5% | +23.3% | -2.0% | -2.1% |
| 4 | **+41.8%** | **+26.6%** | -3.1% | -2.2% |
| 8 | +10.1% | +6.6% | -4.5% | -3.6% |

### 4.2.3 WHERE TO INSERT MoE LAYERS

To study where MoE layers are most effective, we conduct an ablation by selecting different insertion strategies. Specifically, we activate MoE experts in (i) the first 8 layers, (ii) the middle 8 layers, (iii) the last 8 layers, and (iv) every third layer throughout the model. This allows us to compare the impact of placing MoE capacity in shallow, intermediate, deep, or evenly distributed positions. We report results on the Amazon-Arts dataset in Table 6.

We observe clear differences depending on where MoE layers are inserted. Using MoE in the first 8 layers yields the weakest performance, suggesting that early representations are dominated by low-level token processing where additional capacity is less beneficial. Distributing MoE every three layers achieves moderate improvements but still falls short. Placing MoE in the middle 8 layers improves results, but the largest gains come from inserting MoE in the

Table 6: Ablation on where to insert MoE layers.

| MoE Placement | Amazon-Beauty $\Delta(\%)$ | | Industrial $\Delta(\%)$ | |
|---|---|---|---|---|
| | NDCG@10 | HR@10 | NDCG@10 | HR@10 |
| First 8 (baseline) | — | | — | |
| Every 3 | +17.7% | +10.3% | +2.0% | +5.3% |
| Middle 8 | +22.8% | +17.2% | +3.1% | +6.9% |
| Last 8 | **+28.4%** | **+27.6%** | **+9.6%** | **+9.0%** |

last 8 layers (+27.6% HR@10 and +28.4% NDCG@10 over baseline). This indicates that deeper layers (where task-specific semantics and collaborative filtering patterns are most prominent) benefit most from specialized experts, as they directly shape the final ranking representations.

### 4.2.4 STATIC VS. DYNAMIC ROUTING

We find that a switch-style (Fedus et al., 2022) dynamic gating severely degrades recommendation quality, while static token-type routing performs much better (Table 5). The likely reason is that static routing gives each expert a clear, consistent role (language vs. item IDs) so they can specialize without interference. In contrast, dynamic routing mixes assignments across experts, leading to greater entanglement between signals and weaker specialization. This highlights that

Table 7: Impact of static routing.

| Routing Strategy | Amazon-Beauty $\Delta(\%)$ | | Industrial $\Delta(\%)$ | |
|---|---|---|---|---|
| | NDCG@10 | HR@10 | NDCG@10 | HR@10 |
| Static | — | | — | |
| Dynamic | -59.5% | -36.9% | -24.2% | -24.4% |

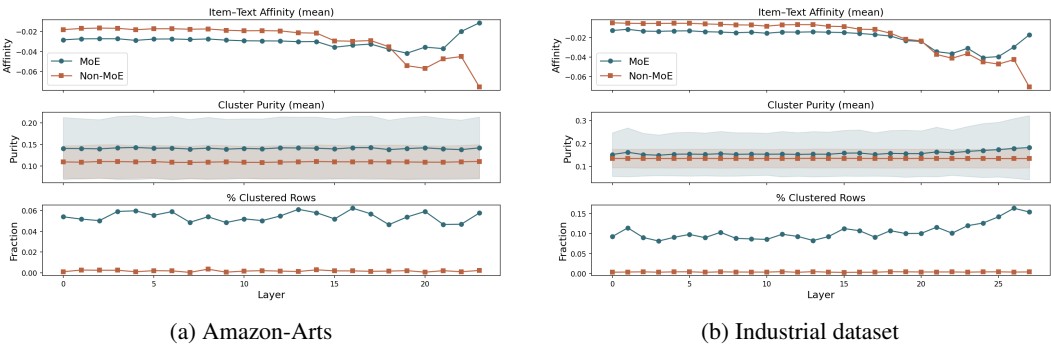

(a) Amazon-Arts  (b) Industrial dataset

Figure 5: FFN key-value memory analysis comparing MoE vs. non-MoE. Each subfigure shows item-text affinity, cluster purity, and fraction of clustered rows across transformer layers.

a fixed separation by token type is not just simpler but also more effective for disentangling language and recommendation signals.

For each layer $\ell$, we report means/medians of $a(w)$ (Equation 2) and $p(w)$ (Equation 3) across rows, and the *clustered fraction* $\mathbb{E}[\mathbf{1}_{\text{cluster}}(w)]$ (Equation 4). In MoE, we compare the item expert. We extract $W_{\text{out}}$ rows, compute top-$k$ similarities to items and text, and summarize per layer and overall. We set $k=20$ and $\tau=0.5$.

### 4.3 FFN KEY-VALUE MEMORY ANALYSIS

The results in Figure 5 show clear differences between MoE and non-MoE models when analyzing FFN neurons as key-value memories. In terms of item-text affinity, both models begin with weak modality preference, but deeper layers of the non-MoE baseline drift toward negative affinity (favoring text), whereas the MoE model maintains more balanced alignment. This indicates that MoE preserves item sensitivity in upper layers, where recommendation decisions are most critical (Table 6).

For cluster purity, MoE consistently yields higher values across layers, meaning that its neurons are more category-specific: when a neuron activates for items, it tends to retrieve items from the same category. Similarly, the fraction of clustered rows (neurons forming coherent category-level clusters) remains low and flat for the non-MoE baseline, is always higher in MoE and rises sharply in the later layers of MoE on the more challenging industrial dataset. Together, these results suggest that MoE separation leads to clearer item-text specialization, higher category purity, and more structured clustering than a vanilla transformer, reinforcing our claim that expert separation enables more interpretable and modular representations of recommendation signals.

## 5 CONCLUSION

We introduced IDIOMoE, a dual-expert continued-pretrained language model that processes text and item data through two specialized experts. Despite its simplicity, IDIOMoE outperforms both classical and recently proposed LLM-based recommendation models. It effectively preserves the pretrained knowledge of the LLM. Our findings highlight the importance of using specialized sub-networks for different modalities, rather than scaling indiscriminately with a single model for all inputs. We view IDIOMoE as a step toward more sustainable and adaptive LLMs for recommendation tasks, and believe this direction is crucial in our efforts to achieve better recommendation performance and interpretability without relying on unnecessarily large models that exhibit diminishing returns.

## 6 ACKNOWLEDGMENTS

The authors would like to thank Zahra Miri for her assistance in preparing the figures.
This work was partially supported by NSF IIS 2347592, 2348169, DBI 2405416, CCF 2348306, CNS 2347617, RISE 2536663.

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

## A  EXTENDED RELATED WORK

### A.1  CLASSIC RECOMMENDATION APPROACHES

Recommender systems have long relied on two complementary paradigms: collaborative filtering (CF) (Yao et al., 2021; Wang et al., 2025a; Li et al., 2022; He & McAuley, 2015) and content-based (CB) methods. CF models exploit user–item interaction patterns, such as ratings or clicks, to learn latent representations of users and items (Koren et al., 2009). This approach is domain-agnostic and often yields high accuracy, but it suffers from well-known *cold-start* problems for new users or items and can exhibit strong popularity bias (Abdollahpouri et al., 2019), over-recommending popular items at the expense of long-tail discovery. CB methods instead leverage explicit item features or descriptions to recommend similar items, which can address item cold-start but ignore collaborative patterns and the "wisdom of the crowd." These methods may produce over-specialized recommendations that limit serendipity.

Hybrid recommenders attempt to combine CF and CB to balance relevance, novelty, and coverage. However, even hybrid systems can be difficult to control with respect to multi-objective goals like fairness, diversity, or novelty without post-hoc re-ranking.

**Sequential and Contextual Models.**  Moving beyond static recommendation, sequential models (Yuan et al., 2018; Zhou et al., 2020; de Souza Pereira Moreira et al., 2021; Hou et al., 2022; 2023a; Wang et al., 2023a) predict a user's next interaction by modeling temporal dependencies in their history. Early neural solutions include GRU4Rec (Hidasi et al., 2015), which applied gated recurrent units to capture sequence dynamics. The introduction of Transformers brought a step-change: SASRec (Kang & McAuley, 2018) was the first to model next-item prediction in an autoregressive fashion using self-attention, improving short-term preference modeling. BERT4Rec (Sun et al., 2019) adapted bidirectional Transformers to better utilize context on both sides of a target position. These architectures form strong baselines in academic and industrial settings, yet they still rely on abstract IDs or dense embeddings, making it hard to integrate external semantic knowledge or to directly optimize multiple objectives beyond accuracy.

Recent work also explores fairness- and diversity-aware training, multi-objective loss formulations, and contextual augmentation, but these methods often require complex pipelines and lack the natural flexibility of a language interface.

### A.2  LARGE LANGUAGE MODELS FOR RECOMMENDATION

The advent of large language models (LLMs) pretrained (Yuan et al., 2020; Xiao et al., 2021; Qiu et al., 2021; Li et al., 2021; Yuan et al., 2021; Shin et al., 2022; Shirkavand et al., 2025a) on massive corpora has opened new opportunities for recommendation. (Zeng et al., 2020; Liu et al., 2023c; Lin et al., 2024b; Yuan et al., 2023; Wang et al., 2024a; Fu et al., 2024) LLMs provide broad world knowledge, reasoning skills, and instruction-following Zhang et al. (2023); Li et al. (2024a); Contal & McGoldrick (2024) abilities that can extend beyond the pattern-matching of traditional recommenders (Zhang et al., 2021b; Muhamed et al., 2021; Cui et al., 2022; Liu et al., 2022; Zhang & Wang, 2023; Wei et al., 2024; Li et al., 2023b; Wang et al., 2023b).

**LLMs as Recommenders.**  A pioneering example is P5 (Geng et al., 2022), which reformulates diverse recommendation tasks into a unified text-to-text format, allowing zero-shot Hou et al. (2024b) and few-shot transfer between tasks such as rating prediction, sequential recommendation, and explanation generation (Bao et al., 2023a; Li et al., 2023c; Yue et al., 2023; Lu et al., 2023; Zhang et al., 2021a; Wu et al., 2024). This unification facilitates integration of multiple modalities, such as textual descriptions or reviews, and enables natural-language queries Liu et al. (2023b); Bao et al. (2023b); Dai et al. (2023); Lin & Zhang (2023); Zhang & Wang (2023); Yang et al. (2023); Carranza et al. (2024); Kieu et al. (2025). However, item representation in such setups is often token-inefficient—especially for large catalogs—because items must be described in text, and off-the-shelf LLMs lack direct exposure to collaborative signals from user–item interactions (Cao et al., 2024). This leads to a mismatch between the LLM's pretrained knowledge and the domain-specific collaborative knowledge needed for effective recommendation.

**Zero-Shot and Prompt-Based Approaches.** Zero-shot prompting (Hou et al., 2024b; Liang et al., 2025) evaluates an LLM as a ranker given a user's history and a set of candidate items in the prompt. Such methods can achieve competitive performance without task-specific training, demonstrating strong generalization, but are sensitive to prompt design, prone to sequence-order biases, and often ignore subtle interaction semantics.

**Fine-Tuning and Alignment.** To address these limitations, fine-tuning methods adapt LLMs to recommendation tasks while preserving language capabilities Ren & Huang (2024); Zhao et al. (2025); Li et al. (2024b); Wang et al. (2021); Shirkavand et al. (2025c). GDM (Cao et al., 2024) introduces auxiliary natural-language training tasks (e.g., masked item modeling, BPR) to inject collaborative patterns. MQL (Zhai et al., 2025) encodes multimodal item attributes (text, images) into a shared quantitative token space, enhancing cold-start and cross-domain performance. RL-based alignment (Lu et al., 2024) further improves controllability by optimizing instruction-following behavior with preference-based rewards, enabling conversational Friedman et al. (2023); Li et al. (2019); Chen et al. (2019); Kemper et al. (2024); Li et al. (2023a); Tang et al. (2025) and constraint-aware recommendation.

**Item ID Integration and Hybrid Representations.** To avoid verbose item descriptions, several works embed item IDs directly into the LLM's vocabulary. CoVE (Zhang et al., 2025) expands the token set with unique item tokens, enabling single-token recommendations and compressed embeddings. CLLM4Rec (Zhu et al., 2024) extends this with both user and item tokens, combining soft and hard prompts to integrate collaborative semantics. These ID-augmented models improve efficiency and accuracy but risk "knowledge entanglement": naive merging of ID and language tokens can cause interference, harming both recommendation accuracy and language fluency.

### A.3 GENERATIVE AND HYBRID RECOMMENDER MODELS

Generative recommenders recast recommendation as a sequence generation task (Yang et al., 2025), unifying retrieval and ranking in one model. HSTU (Zhai et al., 2024) employs a Transformer-based transducer, scaling up to 1.5T parameters and achieving large offline and online gains, while demonstrating NLP-like scaling laws for recommendation. TIGER (Rajput et al., 2023) compresses item vocabularies via multi-code vector quantization. OneRec (Deng et al., 2025b) unifies retrieval and ranking in an encoder–decoder Transformer with sparse Mixture-of-Experts (MoE) Shazeer et al. (2017); Fedus et al. (2022); Ma et al. (2018); Tang et al. (2020); Xu et al. (2024); Zhang et al. (2024); Wang et al. (2024b) for capacity scaling and adds Iterative Preference Optimization for alignment. These approaches offer novelty, explainability, and unified modeling, but require heavy compute and careful fine-tuning strategies to retain collaborative memory.

**Beyond Accuracy.** Extensions like MTGR (Han et al., 2025) integrate hand-crafted features into generative architectures, while others focus on fairness, calibration, and bias mitigation in LLM-based recommenders (Yang et al., 2025). The generative format naturally supports novelty and explanation generation, which can combat popularity bias and improve transparency, but system design remains challenging.

### A.4 MULTIMODAL MoE LLMs

More recently, MoE has been integrated directly into multimodal large language models (MLLMs) and large vision–language models (LVLMs) Bao et al. (2022); Shen et al. (2023); Diao et al. (2025); Deng et al. (2025a); Xiong et al. (2025); Wang et al. (2025b); Xiong et al. (2026); Ganjdanesh et al. (2025); Shirkavand et al. (2025b). MoE-LLaVA (Lin et al., 2024a) introduces a sparse MoE backbone for LLaVA (Liu et al., 2023a)-style LVLMs and proposes a three-stage MoE-tuning strategy that first builds a strong dense LVLM and then converts its feed-forward blocks into experts. The resulting MoE-LLaVA model achieves performance comparable to or better than substantially larger dense LLaVA variants, while activating only a fraction of the parameters per token and reducing visual hallucinations Rawal et al. (2025).

Uni-MoE (Li et al., 2025) targets unified multimodal LLMs that support a broad set of modalities and tasks, applying MoE layers to scale capacity while maintaining a single generalist model. Addressing task interference in instruction-tuned MLLMs, MoME (Mixture of Multimodal Experts) (Xu et al.,

Table 8: Statistics of Amazon datasets used.

| Dataset | Total sequences | Num items |
|---|---|---|
| Games | 42259 | 13839 |
| Instruments | 17112 | 6250 |
| Arts | 22171 | 9416 |
| Sports | 35598 | 18357 |
| Beauty | 22363 | 12101 |
| Toys | 35598 | 11924 |
| Books(23) | 776370 | 495063 |
| Beauty(23) | 729576 | 207649 |
| Toys(23) | 432264 | 162035 |

2024) decomposes the architecture into a Mixture of Vision Experts (MoVE) and a Mixture of Language Experts (MoLE). MoVE aggregates features from multiple vision encoders via an adaptive deformable transformation and an instruction-conditioned router, while MoLE inserts sparsely gated adapter experts into LLM layers.

## B EXPERIMENTS

### B.1 BASELINES

We benchmark **IDIOMoE** against representative methods spanning classic sequence modeling and recent LLM-based recommenders, with an emphasis on baselines that add recommendation capability to LLMs.

**Early sequential modeling.** *GRU4Rec* (Hidasi et al., 2015) pioneers GRU-based session modeling; *SASRec* (Kang & McAuley, 2018) introduces unidirectional self-attention; *BERT4Rec* (Sun et al., 2019) adopts bidirectional masked modeling for sequences.

**Transformer extensions and self-supervision.** *FDSA* (Zhang et al., 2019) enriches feature dependencies within Transformers, and *S3-Rec* (Zhou et al., 2020) pretrains with sequence-aware self-supervision.

**Representation design, multimodality, and framework-style comparatives.** *VQ-Rec* (Hou et al., 2023b) learns discrete item codes via vector quantization; *MissRec* (Wang et al., 2023a) explores multimodal pretraining and transfer; *TIGER* (Rajput et al., 2023) formulates autoregressive retrieval over semantic IDs. Framework baselines that unify text and recommendation include *P5/P5-CID* (Geng et al., 2022; Hua et al., 2023) and its multimodal extension *VIP5* (Geng et al., 2023). *E4SRec* (Li et al., 2023d) targets efficient sequential recommendation with a largely frozen LLM. *ReAT* (Cao et al., 2024) aligns LLMs to recommendation through auxiliary, recommendation-specific tasks. For completeness on small Amazon benchmarks, we also report *CoVE* (Zhang et al., 2025).

**Our reproduced and controlled variants.** To isolate architectural effects under identical capacity, tokenizer, and training budget, we implement three LLM-based variants on the *same backbone* as IDIOMoE: (i) *ID Transformer* (item tokens only); (ii) *Item-ID LLM + text-derived bias* (ID embeddings augmented with text features); and (iii) *Item-LLM* (vocabulary expansion with explicit item text but no MoE). We also reproduce strong non-LLM and hybrid sequential baselines, including *SASRec* (Kang & McAuley, 2018) and *HSTU* (Zhai et al., 2024). Unless stated otherwise, all LLM-based baselines are matched to IDIOMoE in active parameter count and trained with the same token budget, optimizer, sequence length, and schedules.

### B.2 DATASETS

We use public Amazon Dataset: Games, Intruments and Arts (Ni et al., 2019) as well as Sports, Beauty and Toys McAuley et al. (2015). See Table 8 for dataset statistics. We also train and evaluate on our in-house industrial-scale dataset with millions of users and tens of thousands of items.

## B.3 PREPROCESSING

We take the preprocessed version of Games, Arts, and sports from Zhai et al. (2025). We take small Sports, Beauty and Toys from Zhang et al. (2025). We download 2023 amazon variants from the official website Hou et al. (2024a). Following previous work Rajput et al. (2023), we first filter out unpopular users and items with less than five interactions. Then, we create user behavior sequences based on the chronological order. We use chronological leave-last-k splitting per user: last 1 for test, the preceding 1 for validation, and the remainder for training. Item text comes from title and categories. Maximum item history length is 50 items (most recent first). Maximum total token length (items + text) is 1024. We truncate text first, then items if necessary to satisfy the context size. We pad shorter sequences to 1024 with a special pad token; attention masks prevent loss on padded positions. We take the final unpadded position for evaluation.

## B.4 OPTIMIZATION AND EVALUATION

Optimizer is AdamW (betas $(0.9, 0.9999)$, eps $1e-8$, weight decay $1e-2$). We use linear warmup of 3000 iterations, then a cosine decay learning rate schedule. We tune learning rate with a grid search over $\{1e-3, 1e-5, 1e-5\}$ for IDIOMoE and baselines. Training runs with `bfloat16` on NVIDIA A100-80GB. Batch size is 128. We use standard next-token objectives that minimizes the KL divergence between the data distribution and the distribution of the LLM. We report NDCG@10/50, HR@10/50, and MRR. Metrics are computed over the full catalog. We train for 200 epochs on small amazon datasets and for 50 epochs on larger amazon datasets. For text benchmarks we use `lm-eval-harness` Gao et al. (2024). We constrain the output space to the unseen token items for retrieval quality.

## B.5 IDIOMoE DETAILS

1. Experts per FFN block: 2 (ID expert + Text expert).

2. Routing: static token-type routing (ID tokens → ID expert; text tokens → Text expert).

3. Shared components: attention, LayerNorms, positional embeddings.

4. Expert widths: Text expert width = 1. ID expert width = 1 for ablations. Tuned for main tables.

5. Placement: all-layers become MoE for ablations. last-k with 4,8, 16 is tuned for main results.

6. Freezing Policy: For Table 1 experiments (Text analysis) and ablations, LLM backbone is frozen. In other small-scale runs we select the best among: freeze-all, freeze-text-expert-only, and freeze-attention-only. In industrial dataset, we freeze everything and only train the item experts and item embeddings.

7. Factorized Embedding: On amazon datasets, instead of a single embedding table $E \in \mathbb{R}^{N_{\text{items}} \times d}$, we first project to a lower dimensional space and then to the model dimension to reduce embedding parameters $E = W_l \times W_u$ where $W_l \in \mathbb{R}^{N_{\text{items}} \times d_{mid}}$ and $W_u \in \mathbb{R}^{d_{mid} \times d}$.

8. For main results (not ablations and not Table 1), we warm up the item expert with item-only sequences for 20% of epochs, then gradually mix in text tokens with a linear schedule. Ablations with LLM-based models and Table 1 do not use this warm-up to ensure fairness.

## B.6 RESULTS

### B.6.1 PROPRIETARY RESULTS

Table 9 shows the results on our industrial dataset.

We also conduct an additional experiement on our industrial dataset to study the effect of scaling the model using the Qwen 2.5 Qwen et al. (2025) family (0.5B, 1.5B, 3B, 7B). Figure 6 shows the results. We see that recommendation quality improves with LLM size given enough training data, and the gains of IDIOMoE over Item-LLM as the main baseline are persistent across all model scales considered.

Table 9: Results on our industrial dataset.

| Method | Industrial $\Delta(\%)$ | | |
| --- | --- | --- | --- |
| | NDCG@10 | HR@10 | MRR |
| SASRec (baseline) | — | | |
| HSTU | +10.5% | +2.7% | +13.2% |
| ID Transformer | +21.1% | +8.9% | +23.1% |
| Title-LLM | -81.8% | -87.6% | -98.4% |
| Text-Attr LLM | +25.4% | +14.1% | +25.9% |
| Item-LLM | +23.5% | +13.0% | +24.3% |
| IDIOMoE | +27.1% | +16.6% | +31.2% |

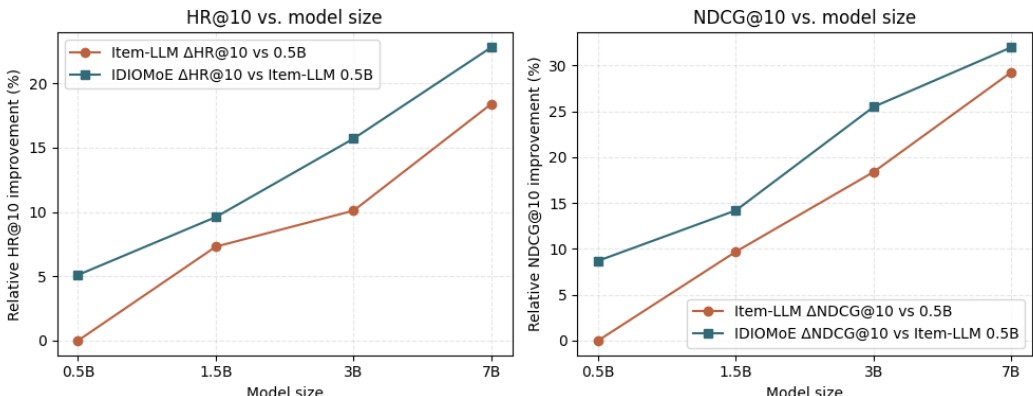

Figure 6: Relative performance on the industrial dataset with Qwen 2.5 backbones of different sizes. All values are reported as relative improvements (%) over the 0.5B baseline Item-LLM.

### B.6.2 SEMANTIC IDS

IDIOMoE is fully compatible with semantic ID schemes for handling new items. We conduct an experiments where we replace raw item IDs with semantic IDs from MQL4GRec (Zhai et al., 2025), showing that IDIOMoE 's gains persist in this setting (Table 10).

We also conducted a cold-start experiment following the steps described in the section 4.3 of TIGER (Rajput et al., 2023), where we remove 5% of test items from the training data and report the test performance overall and over the unseen set items. We set the ratio of unseen items to seen items in the top-k items $\epsilon = 0.1$. Table 11 shows the results, demonstrating that our method extends naturally to standard cold-start mechanisms.

These results demonstrate that our method extends naturally to standard cold-start mechanisms and is compatible with semantic-ID-based handling of new items.

### B.6.3 ATTENTION ANALYSIS ON TEXT-ONLY PROMPTS

We analyze the internal attention behavior of our Item LLM on text-only inputs. We use the same tokenizer and pretrained backbone as the deployed model, run the model on a set of text prompts, and compute summary statistics per layer. We compare (i) IDIOMoE (ii) a freshly loaded pretrained backbone.

For each transformer layer, we average heads, mask padding, and re-normalize per query. We report:

1. previous-token attention, $A[i, i-1]$ averaged over valid positions

2. attention to the first token, $A[:, 0]$

3. the distance profile, $A[i, i-d]$ as a function of offset $d$

Table 10: Performance with Semantic IDs on three Amazon datasets.

| | Arts | | Games | | Instruments | |
|---|---|---|---|---|---|---|
| Method | HR@10 | NDCG@10 | HR@10 | NDCG@10 | HR@10 | NDCG@10 |
| Item-LLM | 0.0946 | 0.0658 | 0.0823 | 0.0481 | 0.0826 | 0.0622 |
| IDIOMoE | 0.1018 | 0.0730 | 0.0880 | 0.0492 | 0.0917 | 0.0686 |

Table 11: Cold-start evaluation following TIGER: 5% of test items are removed from training. We report overall test metrics (All) and metrics restricted to unseen items (Unseen) for three datasets.

| | Arts | | | | Games | | | | Instruments | | | |
|---|---|---|---|---|---|---|---|---|---|---|---|---|
| | All | | Unseen | | All | | Unseen | | All | | Unseen | |
| Method | HR@10 | NDCG@10 | HR@10 | NDCG@10 | HR@10 | NDCG@10 | HR@10 | NDCG@10 | HR@10 | NDCG@10 | HR@10 | NDCG@10 |
| Backbone | 0.0808 | 0.0618 | 0.0569 | 0.0395 | 0.0849 | 0.0534 | 0.0478 | 0.0332 | 0.0642 | 0.0433 | 0.0394 | 0.0249 |
| IDIOMoE | 0.0892 | 0.0643 | 0.0547 | 0.0416 | 0.0941 | 0.0572 | 0.0541 | 0.0422 | 0.0877 | 0.0579 | 0.0580 | 0.0313 |

4. the entropy of the attention distribution over keys per query, averaged over queries.

We also aggregate distance profiles over early/mid/late layer blocks for clarity.

Figures 7 and 8 show that the MoE model and the pretrained backbone exhibit *near-identical* attention patterns on text-only inputs across all layers. Layer-wise previous-token bias, first-token emphasis, and attention entropy overlap almost perfectly, and early/mid/late distance profiles coincide within visual resolution.

This alignment is expected in our setting for two reasons:

1. The MoE architecture modifies the feed-forward pathways, while the backbone self-attention blocks remain architecturally unchanged
2. The text-only inputs do not activate item-specific experts, so the effective computation path closely matches the backbone.

Consequently, attention *structure* (diagonal strength, range of contextual aggregation) remains stable, even though token-level representations downstream of attention can still differ due to MoE expert routing within the MLPs. Under text-only prompts, our fine-tuned Item LLM preserves the backbone's attention geometry. This suggests that improvements from MoE primarily arise in representation and computation within expert MLPs rather than from altering attention allocation.

### B.6.4 Efficiency Results

Our model is evaluated with standard batched inference. It is not restricted to processing a single query at a time. Just like a conventional LLM, IDIOMoE supports multi-query batches with appropriate padding and attention masking, and all of our reported results use evaluation batch sizes larger than 1. Table 12 reports latency and throughput values for both batched training and inference on three sequence lengths, showing that IDIOMoE achieves comparable performance values to the underlying backbone model at various sequence lengths. The MoE modification only changes the FFN sublayers (attention remains shared), so the per-token compute remains similar, and there is no additional online adaptation step at serving time beyond a single forward pass. From Table 12 two trends stand out:

1. Overhead shrinks with sequence length. At short contexts (256 tokens), MoE adds modest training overhead (+6.5% latency, -6.1% tokens/s) and a larger inference overhead (+18.4% latency). As context grows, routing/pack–scatter costs amortize: at 512 tokens the inference overhead drops to +12.5%, and at 1024 tokens it is only +3.8% with no memory increase. Training overhead is similarly small at long sequences ($\leq 0.7\%$ tokens/s at 1024).
2. Memory is neutral. Peak GPU memory is within $\pm 0.5$G of the dense baseline across all settings, and identical at 1024 tokens for both training (29.4G) and inference (4.67G), consistent with activating one expert per token.

IDIOMoE achieves near-parity efficiency at long contexts ($\leq 4\%$ overhead at 1024) and acceptable overheads at short contexts ($\approx 18\%$ at 256), while keeping memory effectively unchanged. In Section 4, we show these costs buy consistent quality gains placing IDIOMoE on a favorable quality–latency Pareto frontier.

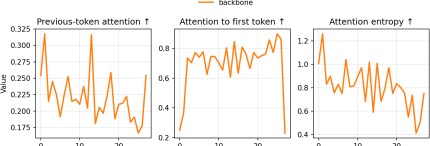 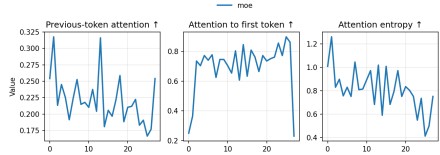

Figure 7: Layer-wise attention metrics on text-only inputs. Left: previous-token attention. Middle: attention to the first token. Right: attention entropy. MoE (blue) and backbone (orange) overlap across layers, indicating preserved attention geometry.

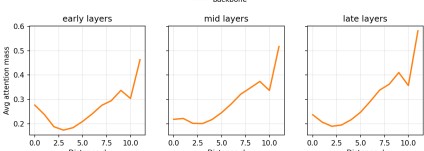 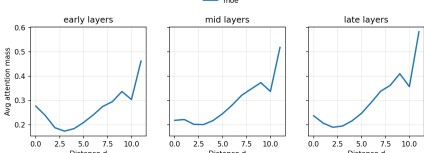

Figure 8: Distance profiles aggregated over early, mid, and late layers. MoE and backbone curves are nearly identical, reflecting similar allocation of attention mass across short-, medium-, and long-range dependencies.

Table 12: Efficiency at batch size $8$ for three sequence lengths with item ratio of $0.2$. $\Delta$ is MoE relative to the dense baseline. Latency is end-to-end per query; throughput is steady-state.

| Seq | Phase | Latency (ms) ↓ | | | Examples/s ↑ | | | Tokens/s ↑ | | | Peak Mem (G) ↓ | | |
|---|---|---|---|---|---|---|---|---|---|---|---|---|---|
| | | Base | MoE | $\Delta$ | Base | MoE | $\Delta$ | Base | MoE | $\Delta$ | Base | MoE | $\Delta$ |
| 256 | Train | 117.86 | 125.53 | +6.5% | 67.88 | 63.73 | −6.1% | 17377.08 | 16314.61 | −6.1% | 10.45 | 10.51 | +0.6% |
| | Infer | 36.13 | 42.78 | +18.4% | 221.44 | 186.99 | −15.6% | 56689.83 | 47870.29 | −15.6% | 2.81 | 2.82 | +0.4% |
| 512 | Train | 180.59 | 186.58 | +3.3% | 44.30 | 42.88 | −3.2% | 22681.76 | 23196.24 | +2.3% | 16.72 | 16.80 | +0.5% |
| | Infer | 49.16 | 55.33 | +12.5% | 162.72 | 144.59 | −11.2% | 83314.50 | 74028.48 | −11.2% | 3.43 | 3.43 | 0.0% |
| 1024 | Train | 323.48 | 323.98 | +0.2% | 24.73 | 24.69 | −0.2% | 25324.61 | 25146.42 | −0.7% | 29.40 | 29.40 | 0.0% |
| | Infer | 92.45 | 95.92 | +3.8% | 86.53 | 83.40 | −3.6% | 88607.00 | 85400.26 | −3.6% | 4.67 | 4.67 | 0.0% |

