# OpenReview forum: "Catalog-Native LLM: Speaking Item-ID dialect with Less Entanglement for Recommendation"
_ICLR.cc/2026/Conference — ICLR 2026 Poster_

### Official Review · Reviewer_jk5P · 2025-10-31

**Soundness:** 3
**Presentation:** 3
**Contribution:** 3
**Rating:** 6
**Confidence:** 4

**Summary:**

This paper proposes a novel architecture for integrating collaborative filtering signals (item interactions) with LLMs. The authors argue that prior works suffer from semantic–collaborative entanglement when item IDs are added directly to LLM vocabularies, and their disentanglement largely improves recommendations while retaining general text reasoning.

**Strengths:**

1. The MoE-based separation of text and item experts is simple yet elegant, showing consistent performance gains on diverse datasets (small/large Amazon and industrial-scale).

2. With only FFN layers modified, this method has the potential to be applied to industrial-scale recommendation systems with low overhead.

3. The paper provides comprehensive ablations of architecture settings like MoE placement, static vs. dynamic routing, and expert shrinkage.

**Weaknesses:**

1. The backbone models are small (0.5B and 1.5B), so it's questionable whether the effectiveness can scale to larger models.

2. The method has limitations on newly-coming items, which require extending the vocabulary and re-training.

3. The idea of assigning different modalities to different experts to reduce modality interference has already been proposed in the domain of VLM / MLLM [1,2]. This paper directly uses this idea but doesn't mention it in related works.

[1]. Li, Yunxin, et al. "Uni-moe: Scaling unified multimodal llms with mixture of experts." TPAMI, 2025.

[2]. Shen, Leyang, et al. "Mome: Mixture of multimodal experts for generalist multimodal large language models." NeurIPS, 2024.

**Questions:**

1. How does IDIOMoE perform in interactive conversational settings (e.g., user dialogue or open-ended preference elicitation)?

2. Could the static routing scheme limit adaptability to unseen token types? How does the model handle cold-start items or dynamically changing catalogs?

3. In lines 121-122, it says previous generative recommendation methods "can forget collaborative structure if not carefully aligned with interaction data.". Can you show more details or evidence?

4. Despite using different experts, your method also integrates the two embeddings after FFN blocks. Can you explain why this is superior to other methods that directly add item tokens to LLM vocabularies (and with some alignment techniques)?

---

> ### Author Response · Authors · 2025-11-23
> **Response to Reviewer jk5P**
>
> We sincerely thank the reviewer for the positive assessment, and the thoughtful feedback. Below we address the concerns raised:
>
> ## W1 (Effectiveness on larger models)
>
> We conducted additional experiements on our industrial dataset to study the effect of scaling the model using the Qwen 2.5 family (0.5B, 1.5B, 3B, 7B) (Table 1 below). These results show that absolute recommendation quality improves with LLM size given enough training data, and the gains of IDIOMoE over Item-LLM as the main baseline are persistent across all model scales considered.
>
> ### Table 1: Relative performance on the industrial dataset with Qwen 2.5 backbones of different sizes. All values are reported as relative improvements (%) over the 0.5B baseline Item-LLM.
> |Model Size|Item-LLM ΔHR@10|Item-LLM ΔNDCG@10|IDIOMoE ΔHR@10|IDIOMoE ΔNDCG@10|
> |-|-|-|-|-|
> |0.5B|+0.0\%|+0.0\%|+5.1\%|+8.7\%|
> |1.5B|+7.3\%|+9.7\%|+9.6\%|+14.2\%|
> |3B|+10.1\%|+18.4\%|+15.7\%|+25.5\%|
> |7B|+18.4\%|+29.3\%|+22.8\%|+32.0\%|
>
> In the paper, we limited public-dataset experiments to <= 1.5B parameters to keep compute comparable across baselines and to support extensive ablations. On the Amazon benchmarks, small models already perform very well. So scaling further there provides little additional insight. Our ongoing work as a follow-up to this paper focuses on training IDIOMoE-style architectures at the 3B+ scale on large industrial data, but the key point from the previous results and added results is that the architectural gains are stable as we increase model size. We have added these results to the revision.
>
>
> ## W2 and Q2 (The method has limitations on newly-coming items, which require extending the vocabulary and re-training.)
>
> Our model is fully compatible with semantic ID schemes for handling new items. We did not include cold-start or semantic ID experiments in the original submission to keep the scope focused on our main contribution: separating information storage for text and item ID modalities when adapting an LLM for recommendation. As per the reviewer's request, we conducted an experiment where we replace raw item IDs with semantic IDs from MQL4GRec[1], showing that (1) IDIOMOE is compatible with Semantic IDs and (2) IDIOMoE’s gains over the main baseline Item-LLM persist in this setting (Table 1 below).
>
> ### Table 1: Performance with Semantic IDs on three Amazon datasets.
> |Method|Arts HR@10|Arts NDCG@10|Games HR@10|Games NDCG@10|Instruments HR@10|Instruments NDCG@10|
> |-|-|-|-|-|-|-|
> |Item-LLM|0.0946|0.0658|0.0823|0.0481|0.0826|0.0622|
> |IDIOMoE|0.1018|0.0730|0.0880|0.0492|0.0917|0.0686|
>
> We also conducted a cold-start experiment following the steps described in the section 4.3 of TIGER[2], where we remove 5\% of test items from the training data and report the test performance overall and over the heldout set items. We set epsilon (the ratio of unseen items to seen items in the top-k) to 0.1. Table 2 below shows the results, demonstrating that our method extends naturally to standard cold-start mechanisms.
>
> ### Table 2: Cold-start evaluation following TIGER. We report overall metrics (All) and metrics on unseen items only (Unseen) for three datasets.
> |Method|Arts HR@10 (All)|Arts NDCG@10 (All)|Arts HR@10 (Unseen)|Arts NDCG@10 (Unseen)|Games HR@10 (All)|Games NDCG@10 (All)|Games HR@10 (Unseen)|Games NDCG@10 (Unseen)|Instruments HR@10 (All)|Instruments NDCG@10 (All)|Instruments HR@10 (Unseen)|Instruments NDCG@10 (Unseen)|
> |-|-|-|-|-|-|-|-|-|-|-|-|-|
> |Item-LLM|0.0808|0.0618|0.0569|0.0395|0.0849|0.0534|0.0478|0.0332|0.0642|0.0433|0.0394|0.0249|
> |IDIOMoE|0.0892|0.0643|0.0547|0.0416|0.0941|0.0572|0.0541|0.0422|0.0877|0.0579|0.0580|0.0313|
>
> These results are of course expected as the data representation format is orthogonal to the contributions of our paper. We have also added these results to the revised paper.
>
> [1] Multimodal Quantitative Language for Generative Recommendation, Zhai et al., ICLR 2025
>
> [2] Recommender Systems with Generative Retrieval, Rajput et al., NeurIPS 2023
>
> ## W3 (VLMs/MLLMs in related works.)
> We thank the reviewer for pointing this out. We agree that MoE-based multimodal LLMs that assign different modalities to different experts are directly relevant prior work and should be discussed. This omission was an oversight. In the revision, we have added a dedicated subsection on Multimodal MoE LLMs to the related work (shorter version in the main paper and a full version in the appendix).

---

> > ### Author Response · Authors · 2025-11-23
> > **Response to Reviewer jk5P (Cont.)**
> >
> > ## Q1 (How does IDIOMoE perform in interactive conversational settings?)
> >
> > IDIOMoE in the format of this paper is trained and evaluated on single-turn recommendation tasks, not full multi-turn conversational sessions. Within this compute budget and data format, we focused on the architectural question (how to specialize FFNs for IDs vs. text—under matched conditions with baselines). Interactive conversational recommendation would require additional training signals (e.g., multi-turn dialogues, diverse prompt formats, synthetic data) and more compute, which we did not include in this submission. Architecturally, IDIOMoE is fully compatible with conversational settings: user history, dialogue turns, and item IDs can be placed in a single sequence, with shared attention over all tokens and the same ID/text expert split. We are currently working on scaling up training with synthetic conversational data and multiple input formats to improve zero-shot and few-shot performance on unseen interactive tasks, including out internal conversational recommendation benchmarks as a follow up to this paper.
> >
> > Regardless below is an example of IDIOMoE (trained with the same data and compute as in Section 4.1.2) on a conversational recommendation prompt format that does not appear in the training set, with content anonymized for double-blind review. The model can follow the instruction and produce recommendations. This instruction-following behavior is one of the main motivations for adapting an LLM to recommendation.
> >
> > **Prompt (unseen format):**
> > *Recommend some items from genre X (a music-related genre in our dataset) to me.*
> >
> > **Response (truncated):**
> > *$\texttt{<|it-38902|>}$ title: YYY, Description: YYY is a fun and educational ITEM that helps you learn about different music genres. You can choose from a variety of genres like pop ...*
> >
> > ## Question 3 (details on previous generative recommendation methods "can forget collaborative structure if not carefully aligned with interaction data.")
> > We agree that the phrase is too strong as written, especially since we do not present a dedicated study of this effect, nor are we claiming that generative recommenders are actually trained on generic language objectives or text-only data. Our intent was only to note that generative recommenders must ensure that their downstream or finetuning training objectives remain close to interaction data so that collaborative signals are fully used. There is not much flexibility on how we can format the input data so the model supports different tasks or objectives.  We have replaced that sentence with "These approaches improve novelty, fluency, and explainability, but are resource-intensive and require careful objective and data design to fully exploit collaborative interaction signals." in the revision.
> >
> > ## Question 4 (Can you explain why IDIOMoE is superior to other methods that directly add item tokens to LLM vocabularies (and with some alignment techniques)?)
> > Our method also adds item-ID tokens to the LLM vocabulary: the model generates both item tokens and text tokens in a single sequence. The key difference from "LLM + item tokens" baselines is not at the embedding level, but in how we allocate capacity deeper in the network. In IDIOMoE, item and text tokens:
> >
> > 1- share the same hidden space and the same self-attention layers at every depth (so they always interact)
> >
> > 2- but use different FFN experts: item tokens go through the item expert FFN, text tokens through the text expert FFN, and the outputs are then merged back into a single sequence.
> >
> > Compared to simply expanding the vocabulary and (optionally) aligning item embeddings, this has two advantages:
> >
> > 1- Dedicated capacity for catalog structure. With direct vocab expansion, all tokens—words and item IDs—must share the same FFN parameters, so catalog-specific patterns compete with general language for the same capacity. In IDIOMoE, collaborative and catalog-specific structure is stored primarily in the item expert, while the text expert preserves the pretrained language behavior. This reduces interference and lets us scale item capacity (via the item expert width/shrink) independently of the text side.
> >
> > 2- Better empirical performance under the same backbone. Our ablations compare IDIOMoE directly to baselines that use the same backbone but only add item tokens (and align them) without MoE. On both Amazon datasets and our industrial dataset, IDIOMoE consistently improves over these ID-augmented baselines, suggesting that separating storage for IDs and text in the FFNs, while keeping attention shared, is more effective than treating item IDs as ordinary vocabulary tokens.

---

### Official Review · Reviewer_jsF2 · 2025-10-31

**Soundness:** 4
**Presentation:** 3
**Contribution:** 3
**Rating:** 6
**Confidence:** 4

**Summary:**

This paper aims to integrate item ids into large language models (LLMs) for recommendation systems. The authors propose IDIOMoE, a Mixture-of-Experts architecture that treats item interactions as a distinct dialect within the language space. The core idea is to route item ids to addtional experts (FFN) and text tokens to the original experts, which allows the model to leverage both collaborative filtering signals and natural language understanding without interference. The paper demonstrates that IDIOMoE outperforms text-only and item-only baselines on both public benchmarks and a large proprietary dataset, while maintaining the semantic understanding of the pre-trained LLM. The authors also provide extensive ablations to show that the improvements come from expert specialization and routing, rather than just added parameters.

**Strengths:**

1. **S1:Idea**: The idea of using a Mixture-of-Experts design to separate collaborative filtering from semantic processing is innovative. By assigning dedicated experts for item IDs and text tokens, the model can mitigate interference and preserve the strengths of both modalities.
2. **S2:Rigorous Evaluation**: The paper provides extensive evaluations on both public benchmarks and a large proprietary dataset compared with various advanced baselines, demonstrating the effectiveness of the proposed method. And authors also conduct extensive ablations and analysis to isolate the source of gains, showing that improvements arise from expert specialization and routing, and providing evidences why IDIOMoE can preserve the LLM's utility.

**Weaknesses:**

1. **Scaling of Parameters**: While the paper shows that the proposed method outperforms baselines, it is not clear how the performance scales with LLM's size.
2. **Ablation on MoE Architecture**: As is shown in Table4, MoA and MoT also achieve similar performance to IDIOMoE, but their parameter design on Attention instead of FFN, what's the advantage of using FFN for MoE instead of Attention?
3. **Can IDIOMoE work with Various ID Types?**: As it shown that the proposed method works well with item IDs, can it also work with other types of IDs, like Semantic IDs in TIGER?
4. **Can IDIOMoE Reasoning in recomendation?**: The paper focuses on the integration of item IDs and text tokens, and it would be interesting to see how well IDIOMoE can handle reasoning tasks that require understanding of both item interactions and natural language queries. For example, can it answer questions about user preferences or make recommendations based on complex user queries that involve multiple items and attributes?

**Questions:**

Refer to Weaknesses and I can further raise my Rating if the authors can provide more results.

---

> ### Author Response · Authors · 2025-11-23
> **Response to Reviewer jsF2**
>
> We sincerely thank the reviewer for the positive assessment, and the thoughtful feedback. Below we address the concerns raised:
>
> ## W1 (Scaling of Parameters)
> We conducted additional experiments on our industrial dataset to study the effect of scaling the model using the Qwen 2.5 family (0.5B, 1.5B, 3B, 7B) (Table 1 below). These results show that absolute recommendation quality improves with LLM size given enough training data, and the gains of IDIOMoE over Item-LLM as the main baseline are persistent across all model scales considered.
>
> ### Table 1: Relative performance on the industrial dataset with Qwen 2.5 backbones of different sizes. All values are reported as relative improvements (%) over the 0.5B baseline Item-LLM.
> |Model Size|Item-LLM ΔHR@10|Item-LLM ΔNDCG@10|IDIOMoE ΔHR@10|IDIOMoE ΔNDCG@10|
> |-|-|-|-|-|
> |0.5B|+0.0\%|+0.0\%|+5.1\%|+8.7\%|
> |1.5B|+7.3\%|+9.7\%|+9.6\%|+14.2\%|
> |3B|+10.1\%|+18.4\%|+15.7\%|+25.5\%|
> |7B|+18.4\%|+29.3\%|+22.8\%|+32.0\%|
>
> In the paper, we limited public-dataset experiments to <= 1.5B parameters to keep compute comparable across baselines and to support extensive ablations. On the Amazon benchmarks, small models already perform very well. So scaling further there provides little additional insight. Our ongoing work as a follow-up to this paper focuses on training IDIOMoE-style architectures at the 3B+ scale on large industrial data, but the key point from the previous results and added results is that the architectural gains are stable as we increase model size. We have added these scaling results to the revision.
>
> ## W2 (Ablation on MoE Architecture)
> Although MoA and MoT are competitive on Amazon-Beauty and occasionally match or slightly exceed IDIOMoE there, we emphasize the industrial-scale results as our primary evidence. As discussed in the paper (lines 343-347), the public Amazon datasets are likely to overlap with pretraining corpora, whereas our internal dataset is free from such contamination. On this large, clean setting, the FFN-based MoE of IDIOMoE consistently outperforms MoA/MoT variants. Nonetheless, the pattern we observe might be dataset-dependent. The core idea of IDIOMoE is to treat catalog items as first-class tokens and to separate where information about IDs and text is stored. All three MoE variants we ablate are consistent with this idea. Our choice to place MoE in the FFNs is guided by the stronger and more stable gains we see on the large-scale industrial dataset.  We have clarified this in the revision and add a short discussion comparing the design choices behind MoA/MoT and our FFN-based MoE.
>
> ## W3 (Semantic IDs)
> Our model is fully compatible with semantic ID schemes for handling new items. We did not include cold-start or semantic ID experiments in the original submission to keep the scope focused on our main contribution: separating information storage for text and item ID modalities when adapting an LLM for recommendation. As per the reviewer's request, we conducted an experiment where we replace raw item IDs with semantic IDs from MQL4GRec[1], showing that (1) IDIOMOE is compatible with Semantic IDs and (2) IDIOMoE’s gains over the main baseline Item-LLM persist in this setting (Table 1 below).
>
> ### Table 1: Performance with Semantic IDs on three Amazon datasets.
> |Method|Arts HR@10|Arts NDCG@10|Games HR@10|Games NDCG@10|Instruments HR@10|Instruments NDCG@10|
> |-|-|-|-|-|-|-|
> |Item-LLM|0.0946|0.0658|0.0823|0.0481|0.0826|0.0622|
> |IDIOMoE|0.1018|0.0730|0.0880|0.0492|0.0917|0.0686|
>
> We also conducted a cold-start experiment following the steps described in the section 4.3 of TIGER[2], where we remove 5\% of test items from the training data and report the test performance overall and over the heldout set items. We set epsilon (the ratio of unseen items to seen items in the top-k) to 0.1. Table 2 below shows the results, demonstrating that our method extends naturally to standard cold-start mechanisms.
>
> ### Table 2: Cold-start evaluation following TIGER. We report overall metrics (All) and metrics on unseen items only (Unseen) for three datasets.
> |Method|Arts HR@10 (All)|Arts NDCG@10 (All)|Arts HR@10 (Unseen)|Arts NDCG@10 (Unseen)|Games HR@10 (All)|Games NDCG@10 (All)|Games HR@10 (Unseen)|Games NDCG@10 (Unseen)|Instruments HR@10 (All)|Instruments NDCG@10 (All)|Instruments HR@10 (Unseen)|Instruments NDCG@10 (Unseen)|
> |-|-|-|-|-|-|-|-|-|-|-|-|-|
> |Item-LLM|0.0808|0.0618|0.0569|0.0395|0.0849|0.0534|0.0478|0.0332|0.0642|0.0433|0.0394|0.0249|
> |IDIOMoE|0.0892|0.0643|0.0547|0.0416|0.0941|0.0572|0.0541|0.0422|0.0877|0.0579|0.0580|0.0313|
>
> These results are of course expected as the data representation format is orthogonal to the contributions of our paper. We have also added these results to the revised paper.
>
> [1] Multimodal Quantitative Language for Generative Recommendation, Zhai et al., ICLR 2025
>
> [2] Recommender Systems with Generative Retrieval, Rajput et al., NeurIPS 2023

---

> > ### Author Response · Authors · 2025-11-23
> > **Response to Reviewer jsF2 (Cont.)**
> >
> > ## W4 (Can IDIOMoE help Reasoning in recommendation?)
> >
> > We designed IDIOMoE as a first step to support exactly this type of joint reasoning. User queries, textual attributes, and item-ID tokens are processed in a single sequence with shared attention, so the model can condition recommendations on multiple items and attributes within the same context. The item expert captures collaborative patterns over IDs, while the text expert handles natural language and attribute semantics. Reasoning about "user preferences" in complex queries amounts to combining these signals through the shared attention layers. In this paper, we evaluated two pieces required for such reasoning: (i) recommendation quality, showing that IDIOMoE improves over non-MoE ID+text baselines and the model understands this new task and (ii) general language capability showing that the backbone’s reasoning ability is preserved.
> > These results (preserving natural language understanding and generation while teaching the model a new modality) show us that the same recipe applied on reasoning for LLMs in math/code domains might be helpful in recommendation tasks too. We are actively working on this direction and studying the effect of data quality, format and size. The next steps for us is to scale up the training data and use synthetic data to boost this reasoning and transparent explanation capability.
> >
> > Even with limited training budget and a simple training format, we currently observe that the trained model can respond to queries in formats not seen in training. We provide an anonymized qualitative example (for double-blind compliance) that show the kinds of recommendation rationales the model can generate.
> >
> > **Prompt (unseen format):**
> > *Recommend some items from genre X (a music-related genre in our dataset) to me.*
> >
> > **Response (truncated):**
> > *$\texttt{<|it-38902|>}$ title: YYY, Description: YYY is a fun and educational ITEM that helps you learn about different music genres. You can choose from a variety of genres like pop ...*
> >
> > Of course this has limitations (e.g. sometimes the model hallucinates) due to the scale of model and training data format and size we used for the paper which we believe will be mitigated with larger scale training. But the main point of our paper is the architectural insights aimed at teaching recommendation capability to the model while preserving text understanding and generation. We can build on that to teach reasoning over history and generating transparent expanations.

---

> > ### Comment · Reviewer_jsF2 · 2025-11-28
> >
> > Thank you for the detailed discussion. I am now leaning toward accept.

---

### Official Review · Reviewer_ZkdV · 2025-11-01

**Soundness:** 3
**Presentation:** 2
**Contribution:** 2
**Rating:** 6
**Confidence:** 4

**Summary:**

This paper addresses the key challenge of integrating collaborative filtering signals (item IDs) and natural language understanding in recommendation systems without mutual interference. It proposes IDIOMoE (Item-ID + Natural-language Mixture-of-Experts Language Model), a dual-expert architecture that treats item interaction histories as a "native dialect" within the language space. The core design modifies the Feed Forward Network (FFN) of a pretrained LLM into separate text and item experts, with static token-type gating to route text tokens to the text expert and item-ID tokens to the item expert. Experiments on public Amazon datasets (6 small-scale, 3 large-scale) and a proprietary industrial dataset (hundreds of millions of users) demonstrate that IDIOMoE outperforms classical recommenders (e.g., SASRec, BERT4Rec) and LLM-based baselines (e.g., P5-CID, CoVE) in metrics like NDCG@10, HR@10, and MRR, while preserving the LLM’s natural language competence. The paper’s key contributions include: (1) a disentangled MoE architecture for separating collaborative filtering and semantic processing; (2) robust performance at industrial scale with retained language understanding; (3) extensive ablations verifying gains from expert specialization rather than parameter scaling; (4) FFN key-value memory analysis revealing more interpretable and modular representations via expert disentanglement.

**Strengths:**

## 1. Originality.
- The paper introduces a novel MoE-based disentanglement paradigm for recommendation systems, which is the first to explicitly model item IDs as a "dialect" separate from natural language. This addresses the long-standing "knowledge entanglement" issue in prior LLM-based recommenders (e.g., CoVE, CLLM4Rec) that naively merge ID and text tokens.
- The static token-type gating design is a creative simplification of standard MoE (e.g., Switch Transformers), avoiding the inefficiency and instability of dynamic routing while achieving effective modality separation.

## 2. Significance.
- It establishes a new direction for LLM-recommender fusion by emphasizing "modality-specific specialization" over brute-force parameter scaling, inspiring follow-up work on disentangled multi-modal recommendation.

## 3. Rigorous experimental design.
- The study covers diverse datasets, comprehensive baselines (classical sequential models, LLM-based recommenders, capacity-matched non-MoE variants), and key ablations (expert capacity, MoE layer placement, static vs. dynamic routing).

**Weaknesses:**

## 1. Insufficient analysis of cold-start scenarios.
- Cold-start is a critical challenge for recommenders, but the paper only mentions item textual attributes as a potential solution without quantitative evaluation. For example: How does IDIOMoE perform on new items with only text descriptions (no interaction data)? And can the item expert leverage text-derived signals to mitigate cold-start bias?

## 2. Insufficient verification of recommendation explanation generation capability.
The paper claims that IDIOMoE retains the language ability of LLM to support "transparent explanations", but the experiments only verify the language understanding ability through general language benchmarks such as WikiText NLL and BBH, without conducting quantitative/qualitative evaluations on explanation generation in recommendation scenarios.

**Questions:**

## 1. Questions Related to Technical Mechanisms.
- Dynamic Adjustment of Item Expert Capacity: The paper observes that small-scale datasets (e.g., Amazon-Beauty) perform best with a smaller-capacity item expert (shrink=4), while large-scale industrial datasets require a full-capacity expert (shrink=1). However, it does not propose an adaptive capacity allocation algorithm that can automatically adjust based on data characteristics (e.g., item interaction sparsity, dataset size). Is it feasible to design such an algorithm?

- What is the performance of IDIOMoE on cold-start items (no interaction data, only text attributes)? Please provide quantitative results on cold-start subsets of the Amazon or industrial dataset, comparing with content-based and LLM-based cold-start methods.

## 2. Suggestions Related to Experimental Design.
- Validation of Recommendation Explanation Generation: The abstract claims IDIOMoE preserves LLM capabilities to support "transparent explanations," but experiments only verify general language understanding via metrics like WikiText NLL and BBH benchmarks. To validate this claim, the authors should supplement an "explanation quality evaluation" experiment.

**Details Of Ethics Concerns:**

No concern.

---

> ### Author Response · Authors · 2025-11-22
> **Response to Reviewer ZkdV**
>
> We sincerely thank the reviewer for the positive assessment, and the thoughtful feedback. Below we address the concerns raised:
> ## W1 and Q1.2 (cold-start)
> Our model is fully compatible with semantic ID schemes for handling new items. We did not include cold-start or semantic ID experiments in the original submission to keep the scope focused on our main contribution: separating information storage for text and item ID modalities when adapting an LLM for recommendation. As per the reviewer's request, we conducted an experiment where we replace raw item IDs with semantic IDs from MQL4GRec[1], showing that (1) IDIOMOE is compatible with Semantic IDs and (2) IDIOMoE’s gains over the main baseline Item-LLM persist in this setting (Table 1 below).
>
> ### Table 1: Performance with Semantic IDs on three Amazon datasets.
> |Method|Arts HR@10|Arts NDCG@10|Games HR@10|Games NDCG@10|Instruments HR@10|Instruments NDCG@10|
> |-|-|-|-|-|-|-|
> |Item-LLM|0.0946|0.0658|0.0823|0.0481|0.0826|0.0622|
> |IDIOMoE|0.1018|0.0730|0.0880|0.0492|0.0917|0.0686|
>
> We also conducted a cold-start experiment following the steps described in the section 4.3 of TIGER[2], where we remove 5\% of test items from the training data and report the test performance overall and over the heldout set items. We set epsilon (the ratio of unseen items to seen items in the top-k) to 0.1. Table 2 below shows the results, demonstrating that our method extends naturally to standard cold-start mechanisms.
>
> ### Table 2: Cold-start evaluation following TIGER. We report overall metrics (All) and metrics on unseen items only (Unseen) for three datasets.
> |Method|Arts HR@10 (All)|Arts NDCG@10 (All)|Arts HR@10 (Unseen)|Arts NDCG@10 (Unseen)|Games HR@10 (All)|Games NDCG@10 (All)|Games HR@10 (Unseen)|Games NDCG@10 (Unseen)|Instruments HR@10 (All)|Instruments NDCG@10 (All)|Instruments HR@10 (Unseen)|Instruments NDCG@10 (Unseen)|
> |-|-|-|-|-|-|-|-|-|-|-|-|-|
> |Item-LLM|0.0808|0.0618|0.0569|0.0395|0.0849|0.0534|0.0478|0.0332|0.0642|0.0433|0.0394|0.0249|
> |IDIOMoE|0.0892|0.0643|0.0547|0.0416|0.0941|0.0572|0.0541|0.0422|0.0877|0.0579|0.0580|0.0313|
>
> These results are of course expected as the data representation format is orthogonal to the contributions of our paper. We have added these results to the revision.
>
> [1]Multimodal Quantitative Language for Generative Recommendation,Zhai et al.,ICLR 2025
> [2]Recommender Systems with Generative Retrieval,Rajput et al.,NeurIPS 2023
>
> ## W2 and Q2 (Recommendation Explanation)
> Our mentioning of "transparent explanations"comes from the fact that if the model does not have human language understanding and generation capability, it will naturally fail to explain recommendation choices too.
>
> We agree that a more direct evaluation of explanations would be useful. However, quantitatively measuring "transparent recommendation explanations" is not straightforward and we are not aware of a standard public benchmark for this. If the reviewer has a specific public benchmark in mind, we would be happy to evaluate on it. We chose to focus the quantitative experiments on the architectural question only: can we specialize FFNs for IDs vs. text while keeping the LLM’s general language ability? We are currently working on a follow-up ot this paper which shows the effect of synthetic data and large scale training in unlocking the model's text understanding to generate exaplanations for recommendation.
>
> Moreover, in experiments on our large internal dataset, we see that the model can respond to formats it has not seen during training. We have provided an anonymized qualitative example ( for double-blind compliance) that show a glimpse of the model's ability:
>
> **Prompt (unseen format):**
> *Recommend some items from genre X (a music-related genre in our dataset) to me.*
>
> **Response (truncated):**
> *$\texttt{<|it-38902|>}$ title: YYY, Description: YYY is a fun and educational ITEM that helps you learn about different music genres. You can choose from a variety of genres like pop ...*
>
> ## Q1.1 (Item Expert Capacity)
> The goal of our ablation in Section Table 5 is not to propose a new adaptive-capacity algorithm, but to show that IDIOMoE exposes a simple, effective capacity knob that makes it usable on both small and large-scale datasets. On Amazon-Beauty, moderate shrink values (2–4) already give large gains over the baseline, while on the industrial dataset shrink=1 works best. In practice, the performance differences among reasonable shrink values are modest, so a fixed choice per domain is enough.
>
> Designing a fully automatic capacity allocator is feasible but orthogonal to our main contribution. A simple classifier could be designed to predict shrink values based on basic data statistics (catalog size and avg interactions per item) or we can select an optimal value via a short validation sweep when adapting to a new domain. Our results already demonstrate that IDIOME's static capacity control is enough to cover both sparse small-scale and dense industrial settings.

---

### Official Review · Reviewer_BYeB · 2025-11-04

**Soundness:** 3
**Presentation:** 3
**Contribution:** 3
**Rating:** 6
**Confidence:** 4

**Summary:**

This paper proposes IDIOMoE, an item-ID and natural-language mixture-of-experts transformer that routes ID tokens to a dedicated collaborative expert and text tokens to the frozen pretrained expert, enabling seamless recommendation within a generative LLM while preserving linguistic ability.

**Strengths:**

1. The paper introduces a novel ID/text mixture-of-experts architecture that treats item IDs as a native dialect, effectively disentangling collaborative and semantic signals.

2. Static token-type routing keeps implementation lightweight while preserving the pretrained LLM’s language capabilities.

3. The method is thoroughly evaluated against industrial-scale baselines on datasets, demonstrating consistent gains.

**Weaknesses:**

1. It seems that the rigid ID/text routing cannot jointly model attributes blended in user queries, leaving collaborative signals and side features separated. For exmaple, in queries like “white iPhone”, the tokenizer sends “white” to the Text-Expert and “iPhone" to the Item-Expert. How can the proposed method avoid such effects?

2. How can the proposed model handle new items? It seems that there is no clear zero-shot or cold-start mechanism is provided in the paper.

3. How efficient is the algorithm proposed in the article? From the model's perspective, it can only process one user's query at a time. If there are a large number of users making recommendation requests, can the model be effectively deployed and adapted?

**Questions:**

See Weakness.

---

> ### Author Response · Authors · 2025-11-22
> **Response to Reviewer BYeB**
>
> We sincerely thank the reviewer for the positive assessment, and the thoughtful feedback. Below we address the concerns raised:
>
> ## W1 (The rigid ID/text routing cannot jointly model attributes blended in user query)
> The concern seems to based on a misunderstanding of our static routing mechanism. In our method, only item ID tokens $\texttt{<|it-⋅|>}$ are routed to the item expert. All other tokens, including all words in natural-language queries and all textual attributes, are routed to the text expert. This is stated explicitly in Section 3.2, lines 213–214: *tokens that are item IDs are sent to the item expert. All other tokens go to the text expert.* Thus, in a query such as *white iPhone $\texttt{<|it-3164|>}$*, both *white* and *iPhone* are routed to the text expert, while $\texttt{<|it-3164|>}$ is routed to the item expert. That said, all tokens mutually attend to each other in the shared attention layers, and cross-attribute reasoning (e.g., matching *white* to items whose descriptions or histories imply a white variant of an *iPhone*) occurs through attention mechanism. The MoE separation only affects where information is stored and updated in the FFNs, not which tokens can interact, consistent with the FFN-as-key–value-memory view we use in our expert analysis. Empirically, our architecture does not exhibit the failure mode suggested in the comment: we observe (1) improved recommendation quality over non-MoE ID+text baselines (Tables 1–3 in the paper) and (2) preserved language understanding relative to the pretrained backbone (Figure 3 in the paper), which would be unlikely if blended language attributes could not be modeled jointly with IDs. We have clarified this in Section 3.2 in the revision.
>
> ## W2 (Cold start)
> Our model is fully compatible with semantic ID schemes for handling new items. We did not include cold-start or semantic ID experiments in the original submission to keep the scope focused on our main contribution: separating information storage for text and item ID modalities when adapting an LLM for recommendation. As per the reviewer's request, we conducted an experiment where we replace raw item IDs with semantic IDs from MQL4GRec[1], showing that (1) IDIOMOE is compatible with Semantic IDs and (2) IDIOMoE’s gains over the main baseline Item-LLM persist in this setting (Table 1 below).
>
> ### Table 1: Performance with Semantic IDs on three Amazon datasets.
> |Method|Arts HR@10|Arts NDCG@10|Games HR@10|Games NDCG@10|Instruments HR@10|Instruments NDCG@10|
> |-|-|-|-|-|-|-|
> |Item-LLM|0.0946|0.0658|0.0823|0.0481|0.0826|0.0622|
> |IDIOMoE|0.1018|0.0730|0.0880|0.0492|0.0917|0.0686|
>
> We also conducted a cold-start experiment following the steps described in the section 4.3 of TIGER[2], where we remove 5\% of test items from the training data and report the test performance overall and over the heldout set items. We set epsilon (the ratio of unseen items to seen items in the top-k) to 0.1. Table 2 below shows the results, demonstrating that our method extends naturally to standard cold-start mechanisms.
>
> ### Table 2: Cold-start evaluation following TIGER. We report overall metrics (All) and metrics on unseen items only (Unseen) for three datasets.
> |Method|Arts HR@10 (All)|Arts NDCG@10 (All)|Arts HR@10 (Unseen)|Arts NDCG@10 (Unseen)|Games HR@10 (All)|Games NDCG@10 (All)|Games HR@10 (Unseen)|Games NDCG@10 (Unseen)|Instruments HR@10 (All)|Instruments NDCG@10 (All)|Instruments HR@10 (Unseen)|Instruments NDCG@10 (Unseen)|
> |-|-|-|-|-|-|-|-|-|-|-|-|-|
> |Item-LLM|0.0808|0.0618|0.0569|0.0395|0.0849|0.0534|0.0478|0.0332|0.0642|0.0433|0.0394|0.0249|
> |IDIOMoE|0.0892|0.0643|0.0547|0.0416|0.0941|0.0572|0.0541|0.0422|0.0877|0.0579|0.0580|0.0313|
>
>
> These results are of course expected as the data representation format is orthogonal to the contributions of our paper. We have also added these results to the revised paper.
>
> [1] Multimodal Quantitative Language for Generative Recommendation, Zhai et al., ICLR 2025
>
> [2] Recommender Systems with Generative Retrieval, Rajput et al., NeurIPS 2023
>
> ## W3 (How efficient is the algorithm proposed in the article?)
> Our model is evaluated with standard batched inference. It is not restricted to processing a single query at a time. Just like a conventional LLM, IDIOMoE supports multi-query batches with appropriate padding and attention masking, and all of our reported results use evaluation batch sizes >> 1. Appendix Table 10 reports latency and throughput values for both batched training and inference, showing that IDIOMoE achieves comparable performance values to the underlying backbone model at various sequence lengths. The MoE modification only changes the FFN sublayers (attention remains shared), so the per-token compute remains similar, and there is no additional online adaptation step at serving time beyond a single forward pass. We have clarified this in the revision and explicitly referred to the latency/throughput table in the appendix.

---

### Meta-Review · Area_Chair_61wF · 2026-01-07

**Summary:**

Reviewers were broadly positive about the core idea (static ID/text MoE disentanglement) and the empirical evidence, but the decision hinged on whether IDIOMoE is complete and credible for real recommender deployment, specifically:
- Cold-start problem: whether IDIOMoE can handle unseen items and whether a cold-start protocol was evaluated.
- Practical efficiency: whether inference is deployable at scale (batching/throughput), and whether MoE introduces meaningful overhead.
- Scalability: whether gains persist for larger LLM backbones.
- Over-claims: especially “transparent explanations” and conversational/interactive recommendation, plus missing related work on multimodal MoE.

**Reviewer Concerns:**

Addressed by the rebuttal (sufficiently for acceptance):
- blended attributes (BYeB): Authors clarified that only FFN storage is separated while attention is shared, enabling cross-token interactions;
- Cold-start (All reviewers): Added semantic ID results and a TIGER-style cold-start evaluation showing gains persist, arguing compatibility with semantic IDs and standard cold-start mechanisms.
- Efficiency (BYeB): Clarified standard batched inference and referenced latency/throughput tables; argued MoE changes only FFN, so per-token compute is similar and there is no extra online adaptation.
- Scaling to larger backbones (jsF2, jk5P): Added scaling experiments on an industrial dataset with Qwen2.5 (0.5B–7B) indicating gains persist across sizes.

Still outstanding / partially addressed (should be noted as limitations or future work):
- Overclaims (ZkdV): Still no standard quantitative evaluation of “transparent explanations”; rebuttal argues benchmarks are lacking and provides limited qualitative evidence. This remains a claim-evidence gap unless reframed.
- Novelty positioning vs multimodal MoE prior work (jk5P): Authors acknowledge omission and add related-work discussion, but this is more positioning cleanup.

**Reviewer Scores:**

Reviewer BYeB (3/3/3)
Likely change: No score change.
Rating: stays 6 or change to 8.
Rationale: Their main concerns (routing, cold-start, efficiency) were directly addressed; they were already positive.

Reviewer ZkdV (3/2/2)
Likely change:
Contribution: 2 → 3 (most likely)
Presentation: stays 2
Soundness: stays 3
Rating: stays 6 or change to 8.
Rationale: Cold-start is now backed by experiments; explanation evaluation remains weak.

Reviewer jsF2 (4, 3, 3)
Likely change: No score change (already strong).
Rating: stays 6 or change to 8.
Rationale: Scaling/semantic IDs/MoE-architecture questions were answered.

Reviewer jk5P (3/3/3)
Likely change: No score change.
Rating: stays 6 or change to 8.
Rationale: Scaling and semantic IDs/cold-start were addressed; related work omission fixed; interactive setting remains future work.

---

### Decision · Program_Chairs · 2026-01-26

Accept (Poster)